# Focal seizures are organized by feedback between neural activity and ion concentration changes

**Damiano Gentiletti[1], Marco de Curtis[2], Vadym Gnatkovsky[2,3], Piotr Suffczynski[1]\***

[1]Department of Biomedical Physics, Faculty of Physics, University of Warsaw, Warsaw, Poland; [2]Epilepsy Unit, Istituto Neurologico Carlo Besta, Milan, Italy; [3]Department of Epileptology, University Hospital Bonn, Bonn, Germany

**Abstract** Human and animal EEG data demonstrate that focal seizures start with low-voltage fast activity, evolve into rhythmic burst discharges and are followed by a period of suppressed background activity. This suggests that processes with dynamics in the range of tens of seconds govern focal seizure evolution. We investigate the processes associated with seizure dynamics by complementing the Hodgkin-Huxley mathematical model with the physical laws that dictate ion movement and maintain ionic gradients. Our biophysically realistic computational model closely replicates the electrographic pattern of a typical human focal seizure characterized by low voltage fast activity onset, tonic phase, clonic phase and postictal suppression. Our study demonstrates, for the first time in silico, the potential mechanism of seizure initiation by inhibitory interneurons via the initial build-up of extracellular $K^+$ due to intense interneuronal spiking. The model also identifies ionic mechanisms that may underlie a key feature in seizure dynamics, that is, progressive slowing down of ictal discharges towards the end of seizure. Our model prediction of specific scaling of inter-burst intervals is confirmed by seizure data recorded in the whole guinea pig brain in vitro and in humans, suggesting that the observed termination pattern may hold across different species. Our results emphasize ionic dynamics as elementary processes behind seizure generation and indicate targets for new therapeutic strategies.

**\*For correspondence:** suffa@fuw.edu.pl

**Competing interest:** The authors declare that no competing interests exist.

## Editor's evaluation

In this manuscript the authors build a small-scale biophysically realistic network model to study seizure dynamics which incorporates Hodgkin Huxley mechanisms and ion dynamics. The model enhances our understanding of the mechanisms underlying the evolution and termination of focal seizures. In particular it demonstrates that intense activation of inhibitory interneurons, by driving changes in transmembrane ion dynamics are a possible mechanism for driving the initiation and prolongation of seizures.

## Introduction

Focal seizure patterns recorded with intracranial and intracerebral electrodes in patients submitted to presurgical evaluation often consist of distinct phases (*Franaszczuk et al., 1998*; *Spencer et al., 1992*; *Velascol et al., 2000*). A frequently observed onset pattern in patients with temporal lobe epilepsy (TLE) is characterized by low-voltage fast (LVF) activity in the gamma range (30–80 Hz) (*Avoli et al., 2016*; *de Curtis and Gnatkovsky, 2009*; *Lagarde et al., 2019*; *Perucca et al., 2014*). LVF seizure onset is followed by the recruitment of irregular spiking behavior which evolves into periodic burst discharges that gradually decrease in frequency and suddenly cease. Seizures are often followed

by a period of reduced EEG amplitude known as postictal EEG suppression. The traditional view on epileptic seizures is that they result from an imbalance of synaptic excitation and inhibition (*Bradford, 1995*; *Bragin et al., 2009*). It is unclear how this concept may account for the electroencephalographic complexity of TLE seizures and their characteristic progression from one phase to the next. The findings of several studies have not confirmed the role of synaptic interaction in seizure generation or progression. It has been shown that blocking synaptic transmission via a low $Ca^{2+}$ solution led to the development of synchronized seizure-like events (SLE) in hippocampal CA1 slices (*Jefferys and Haas, 1982*; *Yaari et al., 1983*) and in the intact hippocampus (*Feng and Durand, 2003*). Moreover, the synchronized epileptiform activity can be recorded across two hippocampal regions separated by a mechanical lesion, without the involvement of electrochemical synaptic communication (*Lian et al., 2001*). Finally, in photosensitive baboons, light-induced neocortical seizure discharges were accompanied by depletion of extracellular $Ca^{2+}$ to levels incompatible with the chemical synaptic transmission (*Pumain et al., 1985*). Additionally, a paradoxical increase in inhibitory cell firing and a decrease in pyramidal cell activity at seizure onset was documented in in vitro rodent slices (*Derchansky et al., 2008*; *Fujiwara-Tsukamoto et al., 2007*; *Lévesque et al., 2016*; *Lillis et al., 2012*; *Ziburkus et al., 2006*), in the in vitro whole guinea pig brain (*Gnatkovsky et al., 2008*; *Uva et al., 2015*), and in human and animal in vivo recordings (*Elahian et al., 2018*; *Grasse et al., 2013*; *Miri et al., 2018*; *Toyoda et al., 2015*; *Truccolo et al., 2011*). The above-mentioned studies suggest that processes at the network level related to changes in synaptic gains cannot be the sole mechanisms that control seizure generation and progression, and that other factors must be involved in the process of ictogenesis.

Although several non-synaptic mechanisms have been proposed to influence abnormal synchronization (*Blauwblomme et al., 2014*; *de Curtis et al., 2018*; *Jefferys et al., 2012*; *Raimondo et al., 2015*), the specific mechanisms responsible for seizure induction, evolution and termination remain unclear. Specifically, the relative contribution of raised extracellular potassium (*Avoli et al., 1996*; *Gnatkovsky et al., 2008*) vs. increased intracellular chloride leading to depolarizing GABA responses (*Alfonsa et al., 2015*; *Lillis et al., 2012*) remains to be established. In the present study, we investigated the possible mechanisms of the LVF seizure pattern using a realistic computational model of the hippocampal network that included activity-dependent ion concentration changes. We based our simulations on the data recorded in the in vitro isolated guinea pig brain preparation because SLE in this model closely resemble human temporal lobe seizures (*de Curtis et al., 2006*; *de Curtis and Gnatkovsky, 2009*) and are initiated by enhanced firing of inhibitory interneurons (*Gnatkovsky et al., 2008*). We used these experimental recordings to guide our in silico study due to the availability of data at the network, cellular, and ionic levels. A computer model showed that simulated seizures initiated via increased interneuron discharges, evolved and terminated autonomously due to activity-dependent ion concentration shifts and homeostatic mechanisms that worked continuously to restore physiological transmembrane ion levels. Our modelling results suggest a link between the seizure termination mechanism and postictal suppression state and predict a specific scaling law of inter-bursting intervals observed at the end of seizures, which was validated experimentally.

## Results

The model consisted of five cells, four pyramidal neurons (PY) and a fast-spiking inhibitory interneuron (IN), arranged as a chain structure (*Figure 1*). The ionic dynamics of $K^+$, $Na^+$, $Ca^{2+}$, and $Cl^-$ were incorporated and activity-dependent changes in their concentrations were computed. Concentration changes in each extracellular or intracellular compartment were dependent on several mechanisms such as active and passive membrane currents, inhibitory synaptic GABAa currents, $Na^+/K^+$-pump, KCC2 cotransporter, glial $K^+$ buffer, $Ca^{2+}$ pump and buffer, radial diffusion, longitudinal diffusion and volume changes. Additionally, we included impermeant anions ($A^-$) with concentration-dependent volume changes and bicarbonate ions ($HCO_3^-$) that contributed to GABAa currents.

### Three phases of an LVF onset SLE

Brief perfusion (3 min) of 50 microM bicuculline in the isolated guinea pig brain transiently reduces GABAergic inhibition to 60–70% and leads to strong interneuron bursting in the absence of principal cells activity (*Gnatkovsky et al., 2008*; *Uva et al., 2015*). Therefore, to initiate a SLE in the model we choose to selectively and transiently enhance discharge of the inhibitory interneuron. In this way,

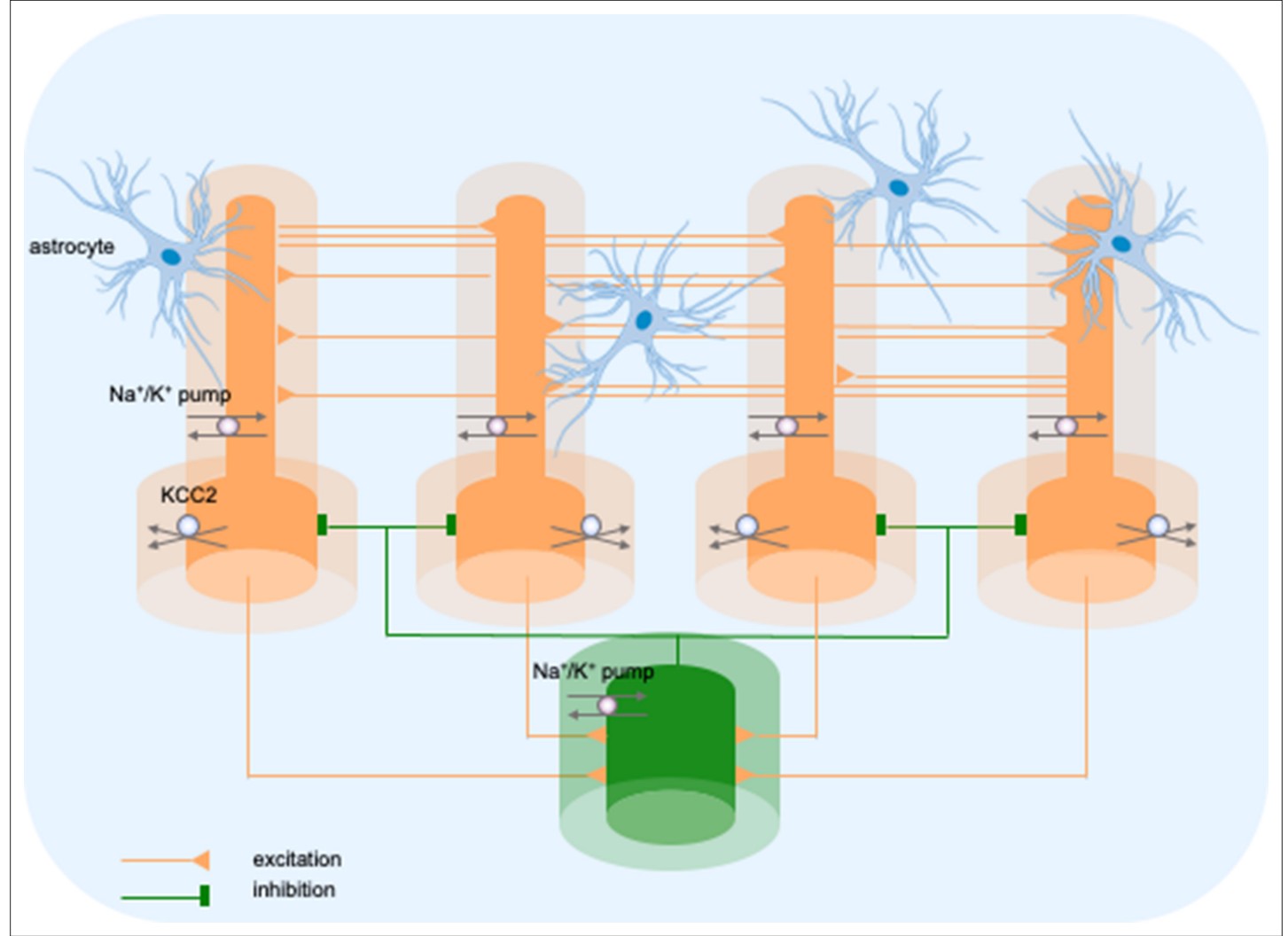

**Figure 1.** Model diagram. The model consisted of four pyramidal cells (orange) and an interneuron (green) linked by excitatory (AMPA) and inhibitory (GABAa) synaptic connections. Each cellular compartment was surrounded by an interstitial compartment. The interstitial space was enclosed in a common bath (blue) which represented the surrounding tissue and vasculature not included in the model. The model included variable intracellular and extracellular ion concentrations computed according to ionic currents flowing across neuronal membranes, longitudinal diffusion between the dendritic and somatic compartments, radial diffusion between neighboring interstitial compartments and diffusion to/from the bath. Additionally, the model included ionic regulation mechanisms: a $Na^+/K^+$-pump, a KCC2 cotransporter and $K^+$ buffering by astrocytes.

we made our model more general and applicable to other experimental data in which paradoxical increase in GABAergic cell firing is observed at seizure onset.

A simulated seizure emerging from normal background activity is shown in *Figure 2*. The SLE was triggered by depolarizing current applied to the IN at second 60 (*Figure 2C*, yellow trace). Strong firing (initial rate of 150 Hz) of the IN (*Figure 2C*) led to small amplitude fast activity in the LFP signal (*Figure 2A* between the 60 and 68 s timestamps) associated with the onset of the simulated SLE. After approximately 10 s, the PY cells began to generate a strong tonic discharge, resulting in an irregular LFP spiking signal with increased amplitude typically associated with the SLE tonic phase (*Figure 2B*; 65–80 s). Approximately 20 s after the onset of the SLE, the cellular firing pattern of PY cells switched from tonic to bursting discharge, leading to LFP oscillations which corresponded to the bursting phase (*Figure 2A*, after 80 s). The three types of PY cell activity are shown in an extended time scale in *Appendix 1—figure 1A*. As the SLE progressed, the burst rate gradually decreased and ictal discharges spontaneously terminated. Postically, the SLE was followed by a period of silence for approximately 90 s which was visible in the LFP signal and in the PY and IN traces (*Figure 2A–C*). After the postictal depression, the background firing reappeared and gradually returned to a baseline

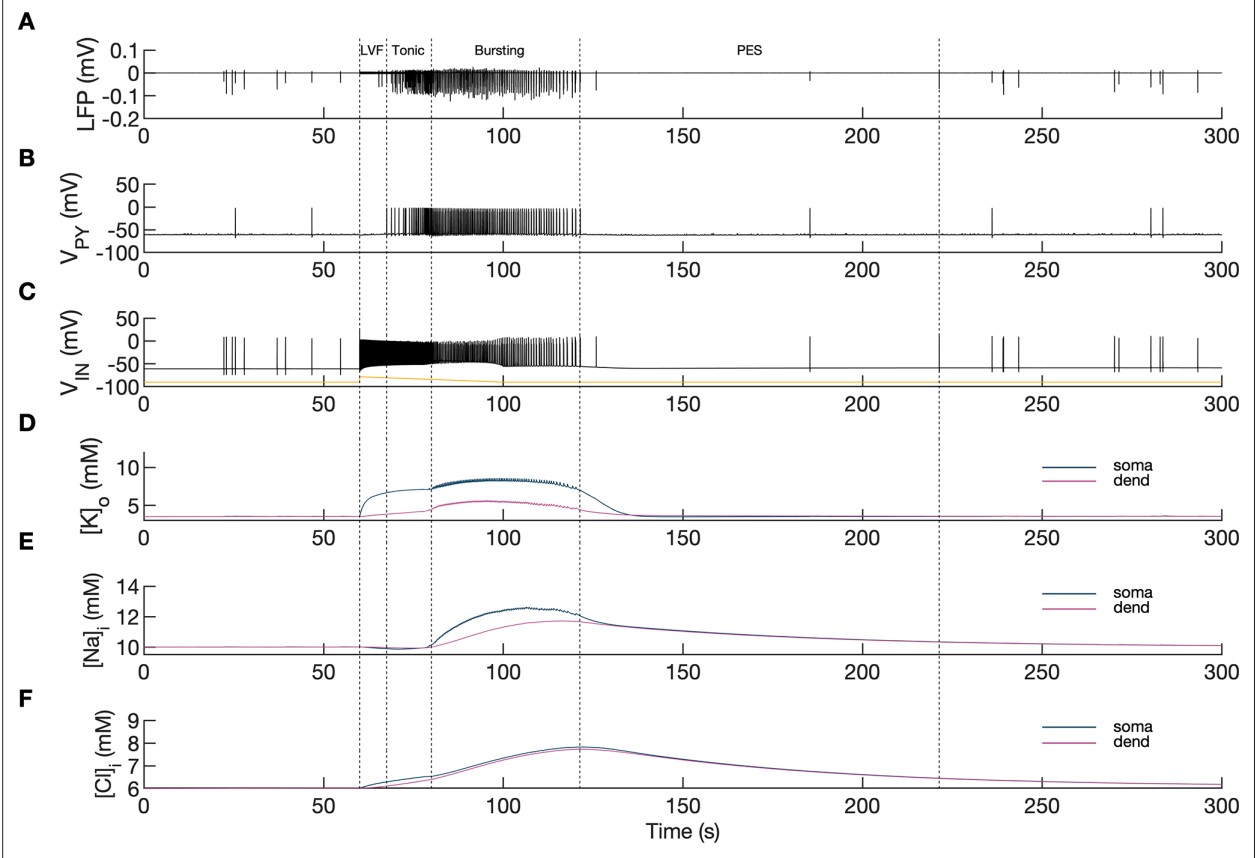

**Figure 2.** Model behavior during an SLE. (**A**) Local field potential (LFP) signal. (**B**) Pyramidal cell (PY) membrane potential. (**C**) Interneuron (IN) membrane potential. (**D**) Extracellular potassium concentration. (**E**) Intracellular sodium concentration. (**F**) Intracellular chloride concentration. In the interictal phase (0–60 s), the model generated irregular background firing and the ion concentrations were at their resting values (**A–F**). The current injected into the interneuron at second 60 (**C**) yellow triggered fast IN spiking (**C**), black which also manifested as low voltage fast (LVF) activity in the LFP signal (**A**) Approximately 10 s after the initiation of the SLE, PY cells initiated tonic firing that subsequently shifted to bursting (**B**) The behavior of the PY cells was reflected in the LFP trace which showed irregular activity and synchronized bursting (**A**) The SLE terminated at approximately second 120 and was followed by postical EEG suppression (PES) (**A–C**). The cellular activity was accompanied by significant ion concentration shifts. Extracellular potassium in the somatic compartment increased sharply and remained elevated throughout the SLE (**D**) dark blue. The $[K^+]_o$ increase in the dendritic compartment was slower and less pronounced (**D**) violet. The intracellular sodium increased gradually toward a plateau (**E**) The intracellular chloride accumulated steadily throughout the SLE (**F**).

The online version of this article includes the following source data for figure 2:

**Source data 1.** Main simulation results.

level. The SLE discharges were accompanied by significant changes in the intracellular and extracellular ionic concentrations. At the onset of the seizure, extracellular potassium concentration ($[K^+]_o$) increased sharply in the somatic compartment, remained elevated throughout the SLE and slightly decreased toward the end of the episode (*Figure 2D*). Intracellular sodium concentration ($[Na^+]_i$) steadily increased during the bursting phase in both somatic and dendritic compartments and reached a plateau around the offset of the SLE (*Figure 2E*). Intracellular chloride concentration ($[Cl^-]_i$) exhibited a gradual increase from the beginning of the SLE and was highest at the end of the paroxysmal firing (*Figure 2F*).

A comparison of the simulation results with the available experimental data is shown in *Figure 3*. In the isolated guinea pig brain, the SLE activity with an LVF onset pattern (*Figure 3A*, top trace) was induced by 3 min arterial application of bicuculine. The transition from preictal to ictal state which occurred at approximately second 10 was associated with a strong discharge of fast-spiking interneurons (*Figure 3A*, third trace) and transient silencing of the PY cells (*Figure 3A*, second trace). Within a few seconds from the initiation of the sustained interneuron discharge, the principal PY cells were recruited first into tonic firing and subsequently into bursting discharges which were visible in the PY

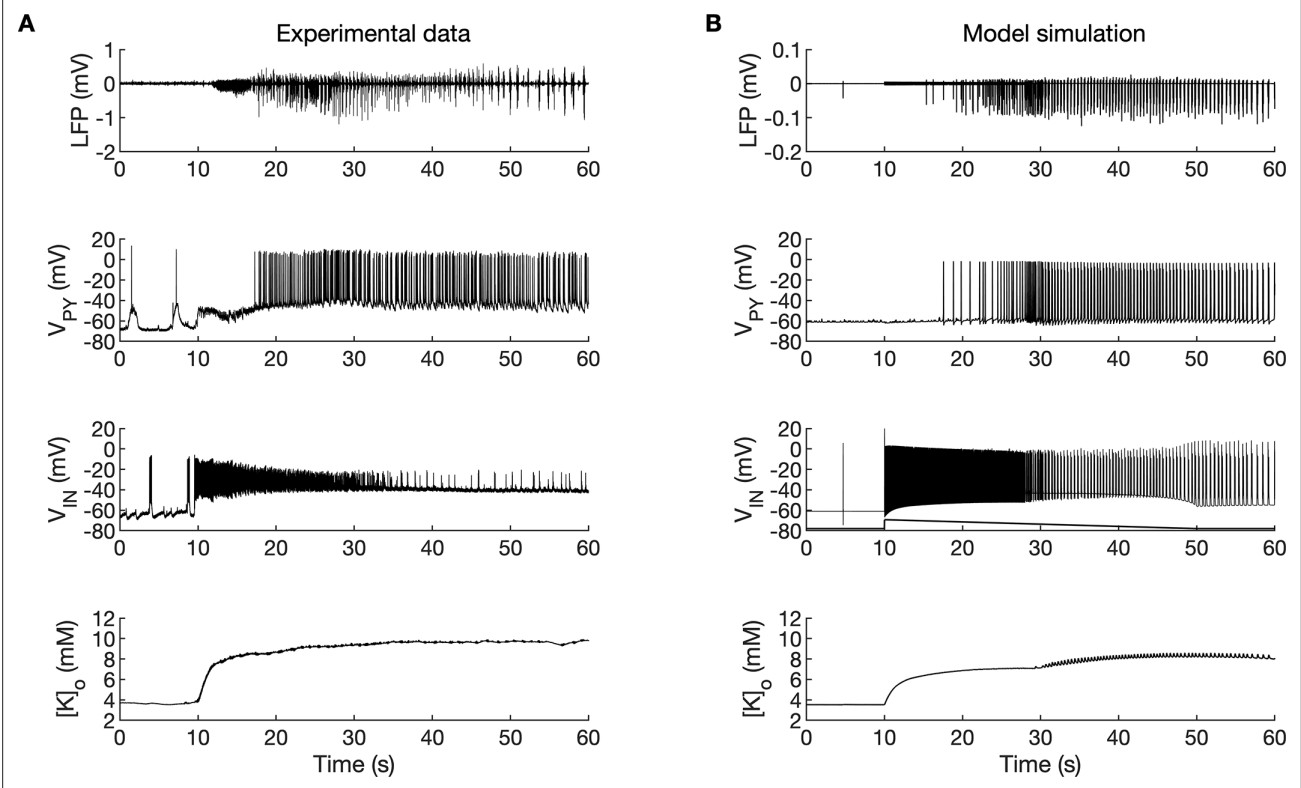

**Figure 3.** A comparison between the experimental data and the model simulation. (**A**) Experimental recordings of a seizure-like event (SLE) in the in vitro isolated whole guinea pig brain preparation (*de Curtis et al., 2006*; *Gnatkovsky et al., 2008*; *Uva et al., 2015*). From top to bottom: LFP signal, intracellular recording of pyramidal cell (PY) and interneuron (IN), extracellular potassium. The onset of the SLE was associated with increased IN firing, silencing PY and low-voltage fast (LVF) activity in the LFP signal. Approximately 10 s after the onset of the SLE, the PY exhibited a tonic and then burst firing behavior. The extracellular potassium increased up to approximately 10 mM at the onset of the SLE and remained elevated afterward. (**B**) The activity patterns in the LFP signal, pyramidal cells, interneuron and $[K^+]_o$ were reproduced accurately by the model. Signals presented in (**A**) were recorded in different experiments. LFP and interneuron data have been published previously (*Gentiletti et al., 2017*; *Gnatkovsky et al., 2008*) while pyramidal cell and $[K^+]_o$ data have never been published before.

The online version of this article includes the following source data for figure 3:

**Source data 1.** Source files for an SLE recordings in the in vitro isolated whole guinea pig brain.

membrane potential and LFP signals. $[K^+]_o$ sharply increased at the onset of the SLE and remained elevated afterward (*Figure 3A*, bottom trace). The in silico results (shown for comparison in the same timescale in *Figure 3B*) replicated the experimental data in many respects including LFP signal characteristics, cellular firing pattern and $[K^+]_o$ time course.

In the simulations, the excitatory and inhibitory synaptic conductances were 'clamped' throughout the entire simulation period, hence, they could not contribute to the progression from one phase to another. Conversely, variations in ion concentrations were expected to affect neuronal excitability. For example, an increase in $[K^+]_o$ reduces the driving force of $K^+$ currents responsible for hyperpolarized resting membrane potential and spike repolarization. $[Cl^-]_i$ accumulation causes depolarizing shift in $E_{GABAa}$ reducing the efficacy of GABAa inhibition. $[Na^+]_i$ and $[K^+]_o$ affect the rate of the $Na^+/K^+$-pump that transports three $Na^+$ ions out of the cell for every two $K^+$ ions pumped into the cell, thus producing an outward current. Accumulation of $[K^+]_o$ and $[Na^+]_i$ increases the pump rate and enhances the hyperpolarizing pump current. Hence, to determine the intrinsic mechanism that modulates excitability, we first investigated the behavior of the model in response to variations in extracellular potassium and intracellular sodium concentrations and next we considered the role of chloride dynamics.

## The effects of $[K^+]_o$ and $[Na^+]_i$ on the network model

Activity-dependent changes in ion concentrations are slow compared to neuronal dynamics, which are relatively fast. To analyze such a system, with slow and fast timescales, it is possible to decouple

the fast variables (e.g. membrane potential) from the slow variables (e.g. ionic concentrations). Accordingly, to analyze the role of $[K^+]_o$ and $[Na^+]_i$ in shaping single-cell and network dynamics, we disabled all mechanisms controlling ionic concentrations and analyzed the behavior of the model for different values of $[K^+]_o$ and $[Na^+]_i$. The values of these variables were modified externally and treated as control parameters. To obtain improved affinity with the reference simulation (*Figure 2*), the chloride concentration in the somatic and dendritic compartments of the PY cells was set to 7 mM, which corresponded to its mean value during the SLE (*Figure 2F*). In the single-cell analysis, all synaptic connections were removed. In the network analysis, all synaptic connections were intact except afferent excitatory input, which was removed from PY cells to eliminate the stochastic component from the analysis; depolarizing current injection was removed from the interneuron. The behavior of a single PY cell was analyzed for different values of extracellular potassium concentration in the dendritic ($[K^+]_{o,dend}$) and somatic compartments ($[K^+]_{o,soma}$). During the analysis, for each fixed value of $[K^+]_{o,dend}$, we performed simulations when sweeping the $[K^+]_{o,soma}$ value from 3 mM to 12 mM (in steps of 0.25 mM) in the forward and backward direction. The initial conditions for each $[K^+]_{o,soma}$ value corresponded to the final states in the previous step. For each $[K^+]_{o,soma}$ step, we simulated 5 s of activity. After a full sweep with all $[K^+]_{o,soma}$ steps in both directions, the $[K^+]_{o,dend}$ was increased and the analysis was repeated. The analysis was performed for the $[K^+]_{o,dend}$ values in the range of 3–6 mM, in steps of 0.5 mM (or 0.25 mM and 0.125 mM if a better resolution was required). We found that the behavior of the single cell and network models was the same for increasing and decreasing steps of $[K^+]_{o,soma}$ with small domains of bistability not larger than one step (0.25 mM) between the activity phases. Analysis of the dynamics of a single isolated cell for a reference value of $[Na^+]_i$ at 10 mM can be found in *Appendix 1—figure 1*. In the network analysis, a full sweep with all $[K^+]_{o,soma}$ and $[K^+]_{o,dend}$ steps (as described above) was performed for three different values of $[Na^+]_{i,soma}$ namely, 10 mM, 11 mM and 12 mM. Following *Figure 2E* we assumed, that corresponding values of $[Na^+]_{i,dend}$ were lower and were 10 mM, 10.5 mM and 11 mM, respectively. The analysis results are shown in *Figure 4A*, in which the various activity patterns, i.e., rest, tonic firing and bursting are color-coded (as shown on the right). A comparison of the network activities for different values of $[Na^+]_i$ in the three graphs in *Figure 4A* demonstrates that the main effect of an increase in $[Na^+]_i$ was a shrinking of the tonic and bursting domains and an expansion of the resting domain due to upregulation of the hyperpolarizing effect of increased $[Na^+]_i$ on the pump current.

## The evolution of the SLE mediated by $[K^+]_o$ and $[Na^+]_i$

In the previous section, the influence of $[K^+]_o$ and $[Na]_i$ on network behavior was examined without accounting for the time factor. Here, we present a fast-slow system analysis approach that considers the time evolution of ionic concentrations, as shown in *Figure 2*. Hence, we assessed the dependence of the evolution of an SLE on externally manipulated changes in $[K^+]_o$ and $[Na]_i$ with fixed concentrations of all other ions. As in *Figure 4A*, the chloride concentration in the somatic and dendritic compartments of the PY cells was set to 7 mM. To schematically describe the extracellular potassium concentration time course from preictal to postictal state (as shown in *Figure 2D*), we identified four distinct stages (as shown in *Figure 4B* top panel): (I) a sharp increase in $[K^+]_{o,soma}$, (II) elevated $[K^+]_{o,soma}$, a slow increase in $[K^+]_{o,dend}$, (III) a slow decrease in $[K^+]_{o,soma}$ and $[K^+]_{o,dend}$, (IV) a decrease in $[K^+]_{o,soma}$ and $[K^+]_{o,dend}$ back to their resting values. Variations in potassium concentrations were accompanied by changes in $[Na^+]_i$ (*Figure 2E*). In the preictal period and during the SLE tonic firing phase, $[Na^+]_i$ in the soma and dendrite was stable, while it increased during the burst firing phase and reached a plateau toward the end of the episode. In the postictal period, $[Na^+]_i$ in both compartments slowly returned to the initial value. The time course of $[Na^+]_i$ approximating intracellular somatic and dendritic sodium evolution is shown in *Figure 4B* (middle panel). The corresponding representative PY cell activity is shown in *Figure 4B* (bottom trace). To distinguish different SLE phases, the PY cell activity pattern was marked using a color-code, as in *Figure 4A*. As $[K^+]_o$ and $[Na^+]_i$ followed their predefined time course, PY cells exhibited transition from rest to tonic firing, progressed into bursting with a slowing-down pattern and eventually returned to the resting state. In the postictal period (after stage IV), the $[Na^+]_i$ remained elevated for more than 40 s (*Figure 4B* middle panel) giving rise to an enhanced hyperpolarizing Na$^+$/K$^+$-pump current that reduced the excitability of the network and contributed to postictal depression, as described in the last paragraph. To further observe the time evolution of the SLE in the $[K^+]_{o,soma}$, $[K^+]_{o,dend}$ parameter space, we superimposed potassium changes on the bifurcation diagrams

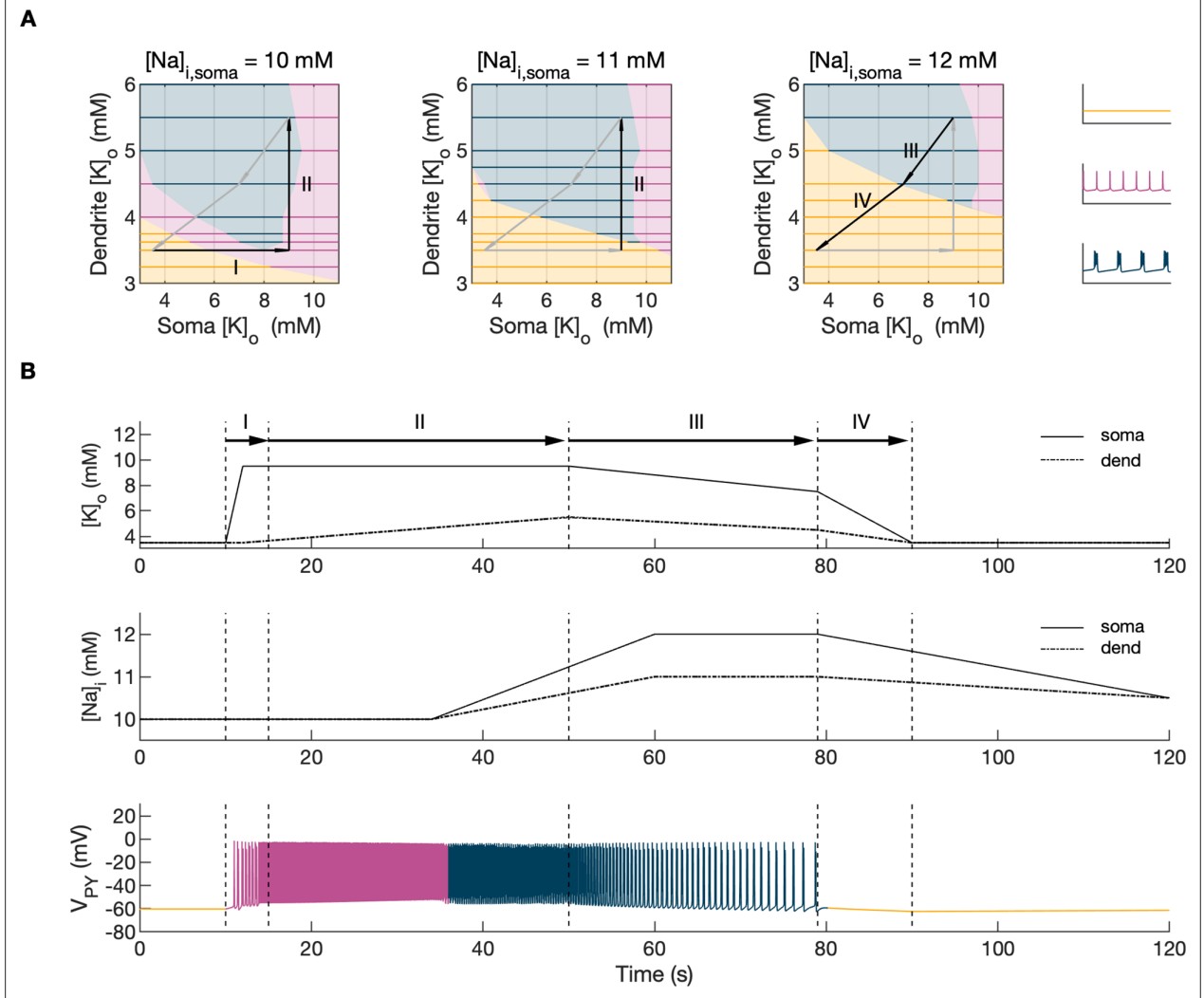

**Figure 4.** Analysis of the model. In the bifurcation analysis extracellular potassium and intracellular sodium concentrations in the PY and IN cells were control parameters. Concentrations of all other ions were fixed at their reference values (except chloride: $[Cl^-]_{i,soma}$, $[Cl^-]_{i,dend}$ equal to 7 mM), all ion accumulation mechanisms were blocked and background input was removed. (**A**) Bifurcation diagrams showing the dependence of the behavior of the model on $[K^+]_{o,dend}$ and $[K^+]_{o,soma}$ for varying values of $[Na^+]_{i,soma}$, $[Na^+]_{i,dend}$. The diagram colors correspond to types of activity shown on the right: rest (yellow), tonic firing (violet) and bursting (dark blue). An increase in $[Na^+]_i$ progressively decreased the domains of tonic firing and bursting and increased the resting domain indicating a general decrease in network excitability. The black and gray arrows correspond to the evolution of $[K^+]_{o,soma}$, $[K^+]_{o,dend}$ during different phases of the SLE, shown in part B. (**B**) A simulation of the model with $[K^+]_{o,soma}$, $[K^+]_{o,dend}$ and $[Na^+]_{i,soma}$, $[Na^+]_{i,dend}$ as the external control parameters, that illustrated the occurrence of transitions between different types of activity during the SLE. The top two panels show the time course of $[K^+]_{o,soma}$, $[K^+]_{o,dend}$ and $[Na^+]_{i,soma}$, $[Na^+]_{i,dend}$ and approximate their evolution during the SLE (*Figure 2*). The third panel shows the resulting PY cell behavior. The parameter evolution is divided into four phases indicated by the arrows denoted as I–IV in part (**A**) and (**B**). Phase I corresponds to a sharp increase in $[K^+]_{o,soma}$ which led to a transition from rest to tonic firing (marked as a black arrow 'I' in the first panel in **A**). Phase II corresponds to a slow increase in $[K^+]_{o,dend}$ which led to a transition from tonic firing to bursting (marked as a black arrow 'II' in the first and second panels in **A**). Phase III represents a period of increased $[Na^+]_{i,soma}$, $[Na^+]_{i,dend}$ and decreasing $[K^+]_{o,soma}$ and $[K^+]_{o,dend}$ which led to the termination of the SLE represented by a black arrow 'III' with its tip in the yellow domain in the third panel in (**A**). Phase IV corresponds to the postictal period with elevated $[Na^+]_i$ and a return of $[K^+]_{o,soma}$, $[K^+]_{o,dend}$ to their baseline values marked as a black arrow 'IV' in the third panel in (**A**).

in *Figure 4A*. The distinct stages of the potassium time course are marked by arrows. The crossing of a color border by an arrow corresponds to a transition between different firing regimes. The initial increase of somatic $[K^+]_o$ (stage I) led to a transition from the resting state to tonic firing (*Figure 4A*, first panel), while a subsequent increase in dendritic $[K^+]_o$ (stage II) led to a transition from tonic firing to bursting (*Figure 4A*, middle panel). A subsequent decrease in somatic and dendritic $[K^+]_o$ (stage III) led to the termination of the SLE, as the cell activity reentered the resting state region (*Figure 4A*,

last panel). After termination of the SLE, $[K^+]_{o,soma}$ and $[K^+]_{o,dend}$ returned to their resting values (stage IV in *Figure 4A*, last panel).

## The role of $[Cl^-]_i$

Chloride accumulation depends on $Cl^-$ influx through chloride leak and GABAa receptor channels, but it is also affected by variations in potassium concentrations mediated via KCC2 cotransport. Hence, chloride and potassium could not be considered as independent control parameters in the fast-slow system analysis approach. Furthermore, visualization of the results with five control parameters ($[K^+]_{o,soma}$, $[K^+]_{o,dend}$, $[Na^+]_{i,soma}$, $[Na^+]_{i,dend}$ and $[Cl^-]_i$) is challenging. Therefore, to evaluate the role of chloride accumulation, we compared the reference model with the model in which chloride dynamics was excluded (*Figure 5*). When considering the role of chloride homeostasis mediated via KCC2, the direction of K–Cl cotransport depends on the $E_{Cl}$ vs $E_K$ or the ratio $[K^+]_i[Cl^-]_i/[K^+]_o[Cl^-]_o$ (*Payne et al., 2003*). If the ratio is greater than 1, KCC2 extrudes $Cl^-$ and $K^+$. If $E_K$ is greater than $E_{Cl}$, the KCC2 flow is reversed and $Cl^-$ and $K^+$ ions are transported into the cell. When chloride accumulation was removed (*Figure 5A*), the reversal potentials of the $Cl^-$ and GABAa currents ($E_{Cl}$ and $E_{GABAa}$) were fixed (blue and light blue lines in *Figure 5A*, second panel). Under such conditions, the firing of the interneuron (*Figure 5A*, third panel) exerted a steady inhibitory influence on the PY cells. Additionally, it increased $[K^+]_o$ above fixed $[Cl^-]_i$ level (*Figure 5A*, fourth panel). As a result, $E_K$ exceeded $E_{Cl}$ and the KCC2 transported $K^+$ and $Cl^-$ into the cells (*Figure 5A*, fifth panel) and reduced the external $K^+$ concentration. All these effects transiently increased the PY cells tonic firing but the bursting SLE phase was not manifested (*Figure 5A*, first and second panel). Conversely, activation of the IN in the reference model, with chloride dynamics intact, led to a typical SLE (*Figure 5B*, top panel). The $[Cl^-]_i$ accumulation was dominated by $Cl^-$ influx through GABAa receptors and to lesser degree by KCC2 cotransport (*Figure 5B*, fifth and bottom panel). It led to enhanced excitability in two ways: (i) by increasing the chloride reversal potential (blue line in *Figure 5B*, second panel) toward the PY membrane potential, reducing the hyperpolarizing chloride leak current; (ii) by increasing the GABAa reversal potential (*Figure 5B*, second panel, light blue) that approached the PY cell membrane potential, reducing the postsynaptic inhibitory current. These changes led to the stronger tonic firing of the PY cells, which contributed to enhanced $[K^+]_o$ accumulation (*Figure 5B*, third panel) leading to the transition into the SLE bursting phase.

## Model predictions

The computer model generated predictions about features that were not explicitly implemented but were consequences of the elementary neurobiological mechanisms used to create the model. These phenomena are described below. The experimental confirmation of features predicted by a model is an essential step in model validation.

## The evolution of the inter-burst interval duration

It is well known that muscle jerking during the clonic phase of a tonic-clonic seizure slows down before ceasing when seizure ends (*Bromfield et al., 2006*). Frequency slowing has also been observed in video sequences (*Kalitzin et al., 2016*) and electrographic counterparts of a seizure revealed by either EEG (*Franaszczuk et al., 1998*; *Schiff et al., 2000*) or EMG (*Conradsen et al., 2013*). In the model, a gradual increase in the interval between ictal bursts (IBI) was visible (*Figures 2A and 3B*). To observe the evolution of the IBI more precisely and identify the scaling pattern, the background noise was removed from the simulation. The absence of excitatory dendritic synaptic input was compensated with a steady depolarizing DC current of 1.85 pA injected into the dendrites of all PY cells. Current intensity was adjusted to preserve the original duration of the SLE as observed in the model with the background noise present. A simulated SLE trace and the detected ictal bursts (short bars) are shown in *Figure 6A* (top panel). The evolution of the IBI is shown below with either a linear (*Figure 6A*, middle panel) or a logarithmic (*Figure 6A*, bottom panel) y-axis. The IBI on the semi-log graph laid on a straight line suggesting an exponential relationship. The evolution of the IBI during an SLE in a whole-brain in vitro preparation (*Figure 6B*) and a human TLE seizure (*Figure 6C*) exhibited the same characteristics. Based on literature, we considered four different scenarios of IBI evolution: linear, exponential, square root, and logarithmic. The fits were evaluated by the root mean square error (RMSE). Exact RMSE values depended on the duration of the analyzed IBI sequence. In the model,

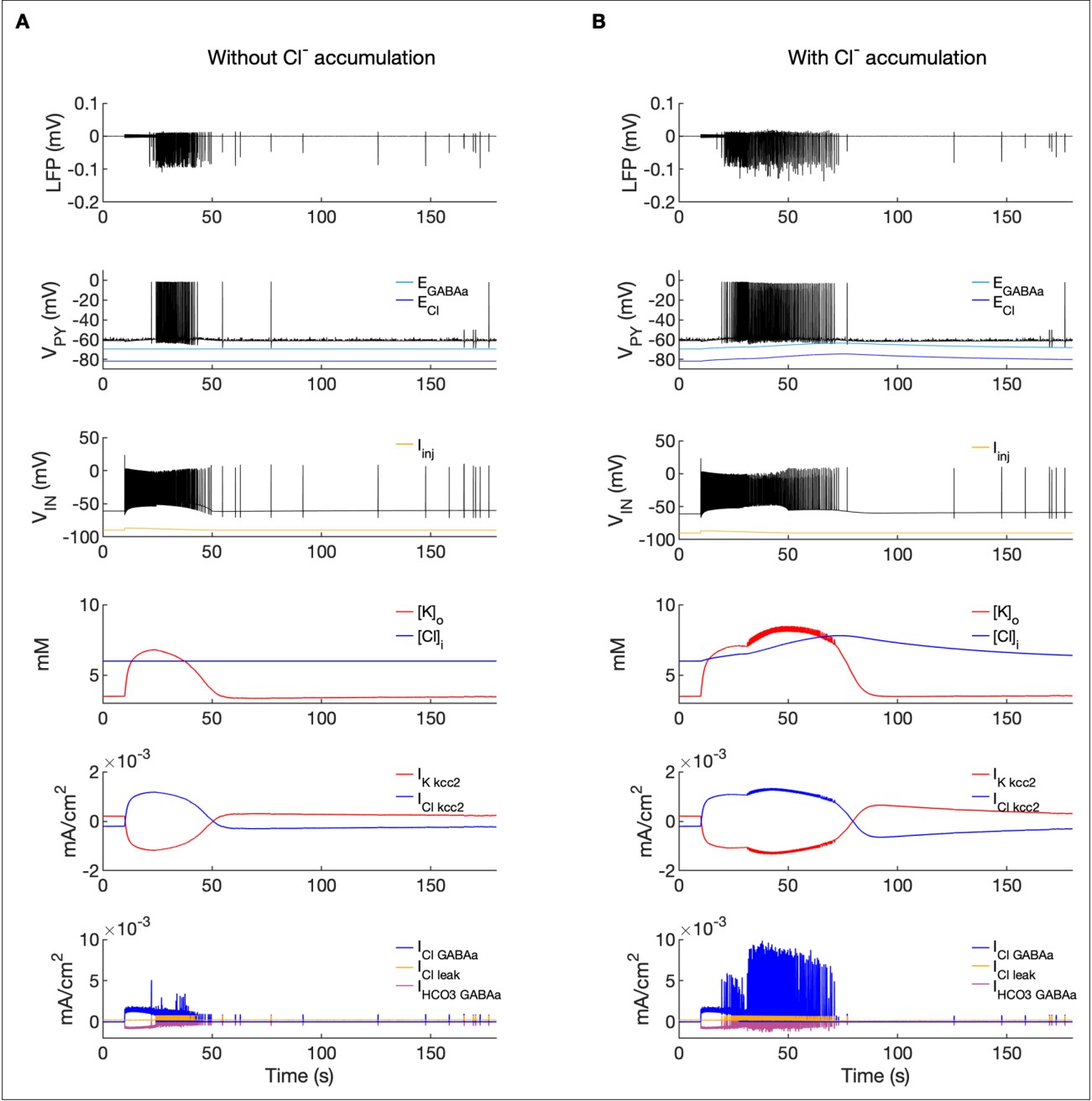

**Figure 5.** A comparison of the model without and with chloride accumulation. The six panels in each column show respectively (from top to bottom): the LFP signal, the PY cell membrane potential, the IN membrane potential, the extracellular potassium concentration and intracellular chloride concentration, the chloride and potassium KCC2 currents in the somatic compartments and the GABAa synaptic currents ($Cl^-$ and $HCO_3^-$) together with the leak chloride current. Additionally, the equilibrium potential of chloride and GABAa are shown in the second panel from the top. (**A**) When the $[Cl^-]_i$ accumulation mechanism was blocked, the chloride concentration was fixed at the reference value (fourth panel, blue). Without chloride accumulation, the PY cell (second panel) fired tonic train of spikes due to transient rise in $[K^+]_o$ (fourth panel, red) mediated by the IN discharge triggered by the current injection (yellow, third panel). Elevated $[K^+]_o$ and fixed $[Cl^-]_i$ promoted $K^+$ influx via KCC2 (fifth panel, red), thus lowering $[K^+]_o$ and further preventing the generation of the full SLE. (**B**) With chloride accumulation, the IN discharge led to an increase in $E_{Cl}$ and $E_{GABAa}$ (second panel, blue and light blue) which reduced the hyperpolarizing $I_{Cl,leak}$ and $I_{GABAa}$ currents and enhanced excitability. The increase in firing rate of the PY cells led to prolonged $[K^+]_o$ accumulation (fourth panel, red) leading to the full SLE.

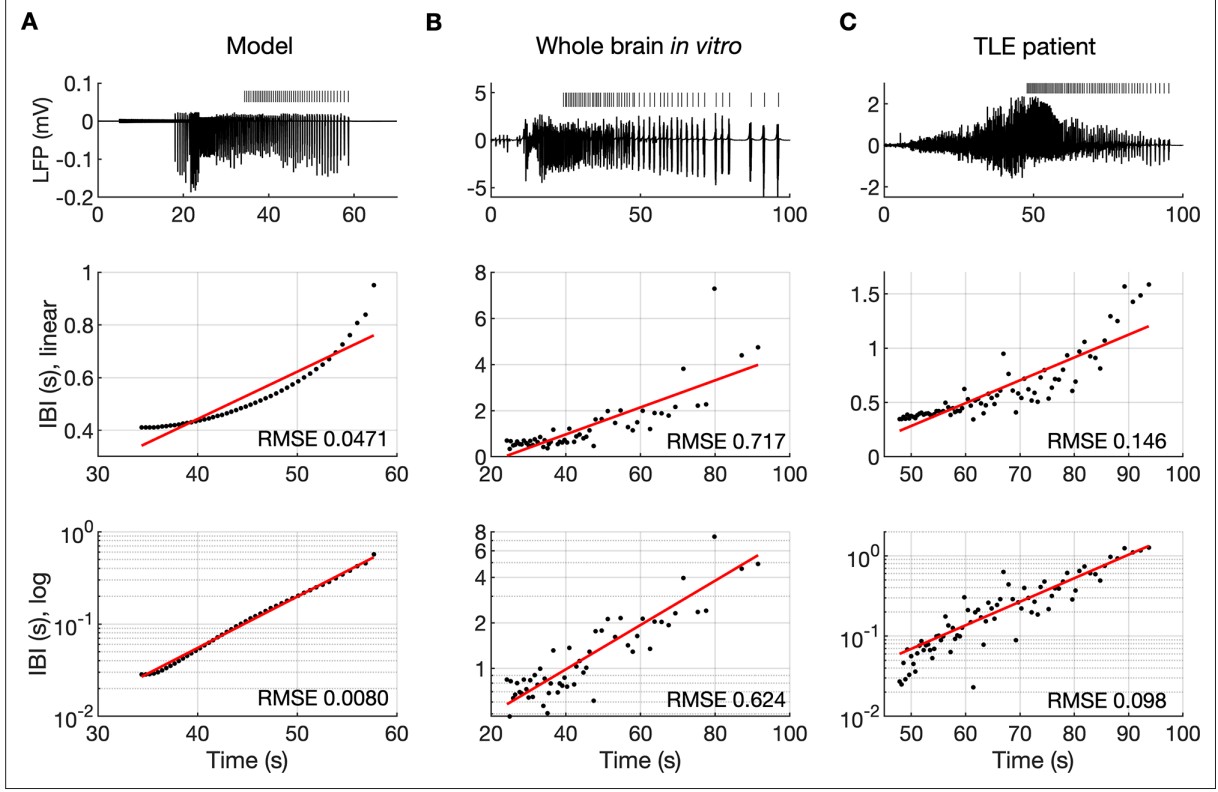

**Figure 6.** The evolution of inter-burst intervals (IBI) in the model and experimental data. (**A**) In the simulation, the background input was removed and compensated with a small depolarizing current injected into the PY cells to preserve the duration of the SLE. A decreasing rate of bursting is visible in the LPF signal and in the detected bursts marked above the trace (top panel). The evolution of the IBI is shown with the y-axis on a linear scale (middle panel) and a log scale (bottom). On a linear y-axis plot, the data appear curved while on a semi-log plot they lay on a straight line, suggesting exponential scaling of the IBI with time. The red line in each plot represents the best fit for the detected IBI; linear function (middle panel) and exponential function providing a linear relationship on a semi-log plot (bottom panel). The root mean square error (RMSE) between the data points and fitted function is shown in each window. The exponential function fit yielded a smaller RMSE compared to the linear, logarithmic or square root fits (see Materials and methods), providing quantitative confirmation that at the end of the simulated SLE, the IBI duration increased exponentially with time. (**B**) The evolution of the IBI during the SLE induced by application of bicuculline in the whole-brain in vitro preparation (*Boido et al., 2014*; *Gnatkovsky et al., 2008*). (**C**) IBI evolution during a seizure recorded with intracerebral electrodes positioned in the temporal lobe in a patient submitted to presurgical evaluation (courtesy of Laura Tassi, Epilepsy Surgery Center, Niguarda Hospital, Milano, Italy). In (**B**) and (**C**), the detected IBI lay on a straight line on the semi-log plot and the exponential fit resulted in a smaller RMSE compared to the linear, logarithmic or square root fits, validating the model prediction of an exponential increase in the IBI at the end of a seizure. Only linear and exponential fits are shown. The results for all considered fits are provided in *Figure 6—source data 1*.

The online version of this article includes the following source data and figure supplement(s) for figure 6:

**Source data 1.** Source files for seizure data used in the analysis of inter-burst interval slowing.

**Figure supplement 1.** Inter-burst interval slowing mediated by ion concentration changes.

**Figure supplement 1—source data 1.** Source files for simulated data used in the analysis of inter-burst interval slowing mediated by ion concentration changes.

RMSE values for logarithmic and exponential fits were often comparable. In the experimental data the exponential fit always had the lowest RMSE. Next, we used the fast-slow system analysis approach, as in *Figure 4*, to demonstrate how separate variations in $[K^+]_o$, $[Na^+]_i$ and $[Cl^-]_i$ affect IBI slowing towards the end of an SLE. Linear decrease in $[K^+]_o$ led to exponential IBI evolution, while linear increase in $[Na^+]_i$ or decrease in $[Cl^-]_i$ led to SLE termination with logarithmic scaling of IBI as determined by RMSE (*Figure 6—figure supplement 1*). Note, that in this figure we simulated a decrease in $[Cl^-]_i$ unlike increasing trend in $[Cl^-]_i$ seen in *Figure 2F*. We observed that a linear increase in $[Cl^-]_i$ didn't lead to IBI slowing and SLE termination, when simulated for up to 20 min (not shown). These results suggest that in the model, slowing of inter-burst interval towards the end of an SLE is mediated by simultaneous changes in $[K^+]_o$ and $[Na^+]_i$.

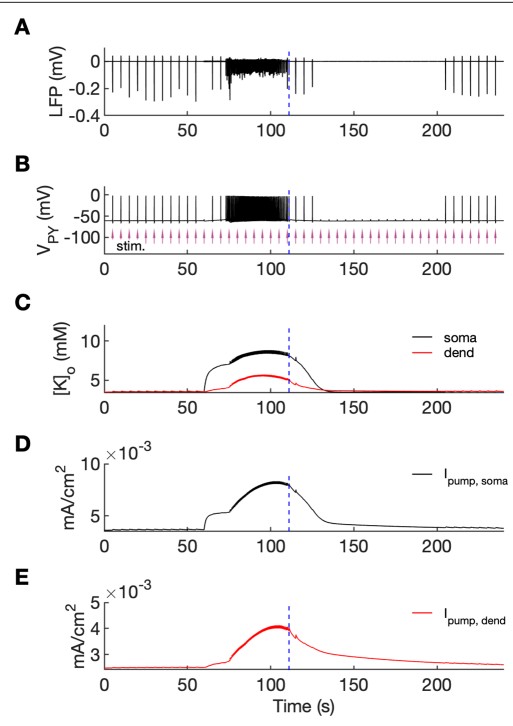

**Figure 7.** An analysis of network excitability in the postictal period. In this figure, the background input was removed from the simulation and compensated with a small depolarizing current injected into the PY cells, as in *Figure 6*. (**A**) The LFP signal. (**B**) The PY cell membrane potential with external periodic stimulation delivered every 5 s, marked by the arrows (violet, stim.). The amplitude of the stimulation was set at just above the threshold for triggering a spike in the interictal period. (**C**) The extracellular potassium in the somatic and dendritic compartments. (**D and E**) The net Na$^+$/K$^+$-pump current in the somatic and dendritic compartment, respectively. The vertical broken line (blue) in all panels marks the SLE offset time without periodic stimulation. Immediately after termination of the SLE, the network was still excitable due to increased [K$^+$]$_o$. Shortly afterward, the excitability decreased due to an increased Na$^+$/K$^+$-pump current that outlasted the increase in [K$^+$]$_o$. Increased $I_{pump}$ and decreased [K$^+$]$_o$ which occurred shortly after the termination of the SLE, led to a postictal period during which the network did not respond to external stimulation for approximately 90 s.

## The postictal period

An additional model prediction concerned the postictal period which is characterized by reduced excitability and firing. As shown in *Figure 2A–C*, after the termination of an SLE there was an approximate 90-s period during which firing was either absent or reduced with respect to the interictal period. The exact duration of the postictal period was difficult to assess, as the gap in firing after the SLE was dependent on background fluctuations. To directly investigate the network excitability, we analyzed the responsiveness of the model to external periodic stimulation (see *Boido et al., 2014*). The background noise was removed and was compensated with a steady depolarizing current, as in *Figure 6*. Stimulation was delivered by activating the excitatory synapses every 5 s at each PY soma (arrows in *Figure 7B*). The amplitude of the excitatory postsynaptic current was set at just above the threshold for triggering the spike in the interictal period (before the timestamp at 60 s). The external stimulation triggered a burst and two single spike responses after the SLE termination (*Figure 7*, vertical broken line) and failed to trigger a suprathreshold response for approximately 90 s afterwards. In line with the PY response pattern, no response to the simulated stimulation was observed in the LFP signal during the postictal suppression period (*Figure 7A*). The high excitability immediately after termination of the SLE (timestamps between seconds 110 and 125) was correlated with elevated [K$^+$]$_o$ which was present shortly after the SLE (*Figure 7C*). The subsequent postictal reduction of excitability was associated with decay of [K$^+$]$_o$ and an increased hyperpolarizing Na$^+$/K$^+$-pump current in somatic and dendritic compartments. The pump current decayed with a slower time constant associated with a gradual clearance of [Na$^+$]$_i$ by the pump (*Figure 7DE*).

## Discussion

The present study aimed to better define the mechanisms underlying focal seizures. The ictal pattern most frequently observed in human and experimental TLE, that is, the LVF onset pattern, exhibits a stereotypical sequence of fast activity, irregular spiking and periodic bursting (as shown in *Figure 3A*, first panel) (*Avoli et al., 2016*; *de Curtis and Avoli, 2016*; *Devinsky et al., 2018*; *Velascol et al., 2000*). We successfully reproduced this pattern in the computer model by transiently increasing the firing of the IN. After this trigger, the simulated SLE phases evolved autonomously. Our study suggests that the various seizure phases and transitions from one phase to another are mediated by feedback mechanisms between neuronal activities, ion concentration changes and ion homeostasis processes. The distinct mechanisms that shape the activities at various seizure stages are discussed below.

## Seizure initiation

There is increasing evidence showing that seizures with LVF onset are initiated by discharges of fast-spiking GABAergic interneurons (*de Curtis and Avoli, 2016*; *de Curtis and Gnatkovsky, 2009*; *Devinsky et al., 2018*). Increased interneuron discharges and decreased PY activity around the time of seizure onset were first evidenced in in vitro and in vivo animal models (*Gnatkovsky et al., 2008*; *Grasse et al., 2013*; *Lévesque et al., 2016*; *Lopantsev and Avoli, 1998*; *Miri et al., 2018*; *Toyoda et al., 2015*; *Ziburkus et al., 2006*). A similar scenario was observed with single-unit recordings performed during intracerebral presurgical monitoring in neocortical and temporal lobe epilepsy patients (*Elahian et al., 2018*; *Truccolo et al., 2011*). Causal relationship between increased inter-neuron firing and ictogenesis may include intracellular chloride accumulation resulting in a shift in $E_{GABAa}$ and/or the elevation of $[K^+]_o$, which leads to subsequent depolarization of PY cells and seizure development (*Magloire et al., 2019b*). The hypothesis suggested by *Jensen and Yaari, 1997* that seizure initiation is related to an elevation in $[K^+]_o$ caused by a strong initial discharge has been tested in computational models with ion concentration changes. In these simulations, seizure-like activity has been induced by DC stimulation of the PY cells alone (*Bazhenov et al., 2004*; *Buchin et al., 2016*; *Kager et al., 2002*), by a brief increase in $[K^+]_o$ (*Fröhlich et al., 2006*), by stimulation of the IN and PY cells (*Y Ho and Truccolo, 2016*) or by varying the extracellular concentration of $K^+$ and $O_2$ (*Wei et al., 2014b*). However, as mentioned above, transitions to spontaneous seizures that begin with LFV pattern were not associated with increased excitatory activity, but with increased firing of inhibitory interneurons. Pyramidal cell – interneuron interplay during SLE was first investigated in the computational model of *Wei et al., 2014a* which mimicked 4-aminopyridine (4-AP) and decreased magnesium in vitro conditions (*Ziburkus et al., 2006*). They showed that when potassium diffusion rate around the IN was lower than around the PY cell, the IN depolarized and increased its activity and eventually entered depolarization block giving way to the strong firing of the PY cell during an SLE. The following in silico tests on seizure initiation by selective involvement of fast-spiking interneurons were conducted by us (*Gentiletti et al., 2017*) and others (*González et al., 2018*). We demonstrated that an increase in interneuron firing triggered a transition to a self-sustained SLE during which both IN and PY cells were active simultaneously. *González et al., 2018* showed that interneuron stimulation by current pulses led to the development of an SLE via a gradual increase in $[K^+]_o$ mediated by the KCC2 cotransporter. Activation of the KCC2 pump was $[Cl^-]_i$ dependent, while $[K^+]_o$ influenced cotransporter time constant.

In the current study, an SLE was initiated by increased IN firing rate in response to depolarizing current injection. We didn't attempt to demonstrate the mechanism leading to increase in IN activity following bicuculline application in the isolated guinea pig brain, as the mechanisms underlying this phenomenon are not fully understood. A decrease in GABAa conductance by bicuculline likely affects interneuron-interneuron inhibition more than interneuron-principal cell inhibition (*Gnatkovsky et al., 2008*). Accordingly, reciprocal release of inhibition between the IN cells (i.e. disinhibition) may lead to preictal interneuronal spikes contributing to increase in extracellular potassium. It would further depolarize interneuronal network and initiate SLE (*de Curtis and Avoli, 2016*; *Figure 4*). In an alternative scenario, increased excitability of interneurons could lead to a transition in a bistable IN network, from asynchronous low firing mode to synchronous high firing rate mode due to small perturbation (*Rich et al., 2020*). In order to fully investigate these effects using a model, one should consider extended interneuronal network with mutual inhibitory interactions. In the current study, we focused on SLE initiation mediated by increased discharge of interneurons, without simulating the underlying processes. It makes the model more general and corresponding to commonly observed paradoxical increase in GABAergic cell firing at the LVF seizure onset. In our present model, the IN was activated by a depolarizing, decreasing current ramp of 40 s. The IN discharge initially led to the silencing of the PY cells, which was correlated with low amplitude fast activity in the LFP signal. The sustained interneuron activity caused a gradual increase in $[Cl^-]_i$ and $[K^+]_o$ in the PY cells. The increase in $[K^+]_o$ produced a positive shift in the $K^+$ reversal potential which led to a reduction in the $K^+$ leak current and membrane depolarization. The accumulation of $[Cl^-]_i$ increased the $Cl^-$ reversal potential and decreased the driving force of the $Cl^-$ ions, which led to a reduction in GABAa IPSC and the $Cl^-$ leak current. An increase in the $E_K$, $E_{Cl}$ and $E_{GABAa}$, and a weakening of the associated hyperpolarizing currents resulted in a gradual depolarization of the PY cells and sustained firing, which was correlated with the tonic SLE phase.

To address the role of potassium and chloride accumulation in seizure initiation, we considered the elevation of $[K^+]_o$ and $[Cl^-]_i$ separately. As shown in *Figure 5*, a selective increase in $[K^+]_o$ with a fixed concentration of $Cl^-$ did not trigger full SLE. When $[Cl^-]_i$ was increased and the concentration of $K^+$ remained fixed at the reference level the PY cells exhibited normal background firing (not shown). This suggests that in our model a change in both $[K^+]_o$ and $[Cl^-]_i$ act in synergy to mediate full SLE. These findings corroborate the results of a study by *Alfonsa et al., 2015* which demonstrated that optogenetic chloride loading of PY cells did not trigger ictal events, while the addition of a subictal dose of 4-AP led to full ictal activity. Our results do not contradict in vitro experimental observations that elevated $[K^+]_o$ alone is sufficient to induce epileptiform activity (*Jensen and Yaari, 1997*; *Traynelis and Dingledine, 1988*). As shown by a bifurcation diagram (*Figure 4A* and *Appendix 1—figure 1*) an increase in $[K^+]_o$ may lead to a transition from a silent state to tonic and burst firing.

Although in our model chloride accumulation increased $E_{GABAa}$ and lowered synaptic inhibition contributing to full-blown SLE, we didn't observe depolarizing GABA responses as seen in some in vitro studies (*Cossart et al., 2005*; *Miles et al., 2012*; *Ellender et al., 2014*). It should be kept in mind that depolarizing GABA responses were found mainly in immature neurons, which have more depolarized $Cl^-$ gradient (*Cherubini et al., 1991*). Another possible explanation of the limited shift in $E_{GABAa}$ in the model may be related to somatic localisation of inhibitory input from the IN. Following the observation that activation of parvalbumin-positive (PV) interneurons was implicated in spontaneous seizures (*Toyoda et al., 2015*), we simulated soma-targeting, PV interneurons but not dendrite-targeting, somatostatin-expressing (SST) interneurons. To see if a more pronounced shift in $E_{GABAa}$ could be observed in SST interneuron mediated dendritic responses, we reduced the size of all model compartments to account for distal dendrites. Under these conditions, strong activation of inhibitory interneuron as in *Kaila et al., 1997* led to biphasic, hyperpolarizing-depolarizing GABAa response shown in *Appendix 1—figure 3*. These results are in agreement with other studies suggesting that $[Cl^-]_i$ can change rapidly and contribute to depolarizing GABAa responses especially in the structures with low volume to GABAa receptor density ratio (*Staley et al., 1995*; *Staley and Proctor, 1999*).

The observation that the development of seizures may be related to an increase in $[K^+]_o$ beyond the physiological values suggests that the modulation of $[K^+]_o$ by the use of $K^+$ chelators is a potential strategy for the control of seizures. In our previous work, we demonstrated that an artificial potassium buffer, which mimicked the function of astrocytes by balancing neuronal $K^+$ release, could reduce neuronal excitability and prevent an SLE (*Suffczynski et al., 2017*).

## The mechanisms of $[K^+]_o$ accumulation

A question arises: what causes $[K^+]_o$ accumulation? The buildup of $Cl^-$ inside the PY neurons during interneuron-induced GABA release results in the activation of the KCC2 cotransporter, which extrudes $Cl^-$ and $K^+$ into the extracellular space. This hypothesis is supported by in vitro experiments which showed that activation of GABAa receptors led to an increase in $[K^+]_o$ and cell depolarization, which were eliminated by the KCC2 inhibitor, furosemide (*Viitanen et al., 2010*). The above-mentioned hypothesis is also consistent with the findings that the application of the KCC2 blockers VU0240551 and bumetanide prevented SLEs during 4-AP application in rat brain slices (*Hamidi and Avoli, 2015*). However, it is not clear whether the ictal activity is consistently based on $K^+$ efflux through KCC2. In the pilocarpine model, the enhancement of KCC2 in principal cortical neurons is associated with a reduction in seizure duration (*Magloire et al., 2019a*). Based on these results, the authors suggested that KCC2 activity does not affect seizure initiation, but influences seizure maintenance during a prolonged period of $Cl^-$ accumulation. The antiepileptic role of enhanced KCC2 activity has also been suggested by *Moore et al., 2018*, who showed that KCC2 potentiation delayed the onset of an SLE after 4-AP application in vitro and reduced the severity of kainate-induced seizures in vivo. When considering the role of KCC2 in $[K^+]_o$ and $[Cl^-]_i$ accumulation, it is necessary to note that the direction and magnitude of KCC2 transport depend on the concentration gradients of $Cl^-$ and $K^+$ (*Kaila et al., 2014*). Under normal conditions, when $[K^+]_o$ is sufficiently controlled by homeostatic mechanisms, GABAergic activity leads to the extrusion of $Cl^-$ and $K^+$ by KCC2. However, an increase in $[K^+]_o$ may reverse the $K^+$–$Cl^-$ cotransport, thus contributing to $[K^+]_o$ buffering rather than accumulation (*Payne, 1997*; *Thompson and Gahwiler, 1989*) The activation of IN in our model led to a rapid increase in $[K^+]_o$ in the narrow extracellular interstitial compartments, while intracellular $Cl^-$ accumulation was more gradual (*Figure 2*). This generated an influx of $K^+$ and $Cl^-$ via KCC2 (*Figure 5B*), which led to $[K^+]_o$

buffering and $[Cl^-]_i$ accumulation. The exclusion of KCC2 involvement in the increase in $[K^+]_o$ suggests that the primary mechanism of $[K^+]_o$ accumulation in our model was due to other processes, such as the outward $K^+$ current that repolarizes action potentials in activated IN and PY cells. This observation is consistent with in vivo experimental evidence that showed a significant local $[K^+]_o$ rise due to increased spiking activity following electrical stimulation of the cat cerebral cortex (*Heinemann and Lux, 1975*).

It is worth noting that in our model KCC2 resumes extruding $Cl^-$ shortly after SLE termination (*Figure 5B*). Accordingly, $[Cl^-]_i$ build up is observed over the whole SLE. Intracellular chloride imaging during SLE induced in vitro by $Mg^{2+}$-free solution showed that $[Cl^-]_i$ started to decline before the end of SLE (*Raimondo et al., 2013*), while during SLE induced in vivo by 4-AP $[Cl^-]_i$ recovery begun instantly after SLE offset (*Sulis Sato et al., 2017*). These dissimilar observations might be related to distinct firing patterns of inhibitory and excitatory neurons in 4-AP and low $Mg^{2+}$ seizure models (*Codadu et al., 2019*), which in turn could lead to different $Cl^-$ and $K^+$ and accumulation patterns.

## The tonic-to-bursting transition

Potassium ions released by interneuron discharges initially diffused to the somatic extracellular compartments of the PY cells and contributed to PY soma depolarization and tonic firing, as described above. The initial fast rise of $[K^+]_o$ in the somatic compartment and slower rise in dendritic segment (*Figure 2D*) was related to the localisation of inhibitory neuron near the PY soma. Subsequently, $K^+$ diffusion from the somatic to dendritic compartments promoted regenerative dendritic spikes in PY cells. In the model, the dendritic conductance of voltage-gated $K^+$ currents associated with spiking was about 10% of the somatic conductance (*Fransen et al., 2002*) hence release of $K^+$ ions into the dendritic interstitial space was smaller than in the somatic compartment. On the other hand, radial diffusion and glial buffering processes had the same efficiency in both compartments maintaining lower dendritic $[K^+]_o$. This model prediction appears to agree with experimental data of simultaneous recordings of $[K^+]_o$ in somatic and dendritic layers during hippocampal seizures in anesthetized rats. During paroxysmal firing induced by electrical stimulation, $[K^+]_o$ in dentate gyrus reached significantly higher levels in cell body layers than in the layers containing dendrites (*Somjen and Giacchino, 1985*). High $[K^+]_o$ in the soma and moderately increased $[K^+]_o$ in dendrites favored burst firing (*Figure 4A* and *Appendix 1—figure 1*) through the reduction of repolarizing $K^+$ currents, the activation of a $Na^+$ persistent current, and a shift from spike after-hyperpolarization toward depolarizing after-potentials. Prolonged depolarization led to the activation of slow M-type $K^+$ conductance (*Appendix 1—figure 2*), which hyperpolarized the PY cells after a series of fast spikes. The bursting mechanism in our model originated from currents used in the original entorhinal cortex cells model (*Fransen et al., 2002*) and was similar to the mechanism in CA1 neurons (*Golomb et al., 2006*).

We note that in our model a transition from resting to tonic and then to bursting activity in PY cells was not critically dependent on perisomatic inhibition and would be also observed if we simulated dendrite-targeting, SST interneurons. As shown by the bifurcation diagram in *Figure 4A* (first panel) an increase in either somatic or dendritic $[K^+]_o$ may lead to a transition from rest to tonic spiking and then bursting. This observation is in agreement with the study showing that optogenetic activation of either PV or SST inhibitory interneurons can trigger SLE (*Yekhlef et al., 2015*).

## Seizure termination

Various mechanisms underlying seizure termination have been suggested (*Lado and Moshé, 2008*; *Zubler et al., 2014*), however, researchers have not yet reached a consensus regarding which one plays a dominant role. In our model, the SLE terminated spontaneously. Following the fast-slow analysis approach (*Fröhlich et al., 2006*), we created a simplified model in which $[K^+]_{o,soma}$, $[K^+]_{o,dend}$ together with $[Na^+]_{i, soma}$ and $[Na^+]_{i, dend}$ were treated as control parameters. Hence, the influence of neuronal activity on ionic variations was removed and the dependence of network activity on $K^+$ and $Na^+$ concentrations was analyzed (*Figure 4A*). When the time course of the concentration changes of these two ions were tuned to reproduce the decrease in $[K^+]_o$ and maintained increased level of $[Na^+]_i$ observed in the late SLE phase, the ictal activity spontaneously terminated (*Figure 4B*). This indicates that SLE cessation in the model can be explained by two coincident factors, namely the decrease in $[K^+]_o$ during stable levels of increased $[Na^+]_i$. An increase in $[Na^+]_i$ led to an increased hyperpolarizing $Na^+/K^+$-pump current, which increased the firing threshold in the neurons. The increased pump activity

also contributed to a progressive decrease in $[K^+]_o$. Potassium repolarization currents increased after each burst and eventually prevented the initiation of a new cycle of the oscillation. The idea that negative feedback between $[Na^+]_i$ accumulation and neuronal firing is responsible for seizure termination was first formulated by *Jensen and Yaari, 1997*. Even though it has not been tested experimentally, this hypothesis is consistent with the observation that the inhibition of $Na^+/K^+$-pump activity occurring during hypoxia prolongs SLE discharges and shortens post-SLE period in hippocampal slices with blocked synaptic transmission (*Haas and Jefferys, 1984*). It was also observed that a decrease in $Na^+$ channel conductance via the antiepileptic drug phenytoin, increased the seizure threshold but prolonged the afterdischarges and seizure durations in the rat kindling model of epilepsy (*Ebert et al., 1997*). In the computational model developed by the Bazhenov team (*Krishnan et al., 2015*; *Krishnan and Bazhenov, 2011*) and *Chizhov et al., 2018*, a progressive increase in $[Na^+]_i$ and activation of the electrogenic $Na^+/K^+$-pump were identified as the primary factor of SLE termination. The seizure termination mechanism in the above-mentioned studies is similar to the mechanism observed in the present study, even though different specifications of neuronal mechanisms, network characteristics and seizure morphologies were used.

It should be also noted that activation of the $Na^+/K^+$-pump by $[Na^+]_i$ is not the only proposed mechanism of seizure termination. An alternative mechanism, also linked to an increase in $[Na^+]_i$, is dependent on the $Na^+$-activated $K^+$ channels (*Igelström, 2013*). Moreover, many other mechanisms such as acidosis (*Ziemann et al., 2008*), the upregulation of inhibitory neurons (*Wen et al., 2015*), glutamate depletion (*Lado and Moshé, 2008*), the depolarization block of neurons mediated by $K^+$ release from astrocytes (*Bragin et al., 1997*), after-hyperpolarization due to $K^+$ channels (*Bazhenov et al., 2004*; *Timofeev and Steriade, 2004*), postburst depression (*Boido et al., 2014*), increased synchrony (*Schindler et al., 2007*) and the release of adenosine *During and Spencer, 1992*; *Uva and de Curtis, 2020* have been suggested to play a role in seizure termination.

The abrupt termination of a seizure across the entire brain (*Salami et al., 2022*) requires long-range communication which may involve thalamocortical interactions (*Aracri et al., 2018*; *Evangelista et al., 2015*), travelling waves (*Martinet et al., 2017*; *Proix et al., 2018*) and ephaptic interactions (*Jefferys, 1995*; *Shivacharan et al., 2019*). Multiple neuromodulatory, ionic, synaptic and neuronal components likely cooperate to terminate a seizure. Further insight into these mechanisms may be obtained by their selective blockage (*Uva and de Curtis, 2020*), the tracking of EEG signal changes as seizure offset approaches (*Boido et al., 2014*; *Saggio et al., 2020*) and from analysis of the duration of postictal suppression (*Payne et al., 2018*).

## Frequency Slowing

The approach of seizure termination is often (but not always) accompanied by an increase in the intervals between successive bursts that form the late seizure phase. *Saggio et al., 2020* analyzed frequency slowing in human focal onset seizures and estimated that approximately 40% exhibited unequivocal discharge slowing down toward the end. Burst frequency slowing was confirmed in a study on the entorhinal cortex of the isolated brain preparation during bicuculline- and 4AP-induced SLE (*Boido et al., 2014*). Our model prediction of the exponential increase in the IBI toward the SLE offset (*Figure 6A*) was confirmed with experimental data (*Figure 6BC*). These findings are also consistent with those of *Bauer et al., 2017*, which demonstrated that inter-burst intervals in focal epilepsy patients were predominantly described by the exponential scaling law. Conversely, it has been suggested that depending on the bifurcation which leads to seizure termination, the burst oscillation frequency can be constant or decrease according to the logarithmic or square root relationship (*Izhikevich, 2000*; *Jirsa et al., 2014*). This theory has not been confirmed by our model and experimental seizure data (*Figure 6*). On the other hand, when specific ion types were varied linearly and led to linearly decreasing membrane current (not shown), logarithmic increase in IBI was observed (*Figure 6—figure supplement 1B,C*). It shows that when the assumption of slow linear membrane current dynamics was satisfied, the IBI evolved according to the bifurcation theory. However, when various processes influenced seizure termination and the current changed non linearly, the IBI slowing deviated from the predicted scaling laws.

In our model, progressive decrease in neuronal excitability related to simultaneous decrease in $[K^+]_o$ and increase in $[Na^+]_i$, was responsible for the IBI slowing toward an SLE end. Alternative explanations for the increasing inter-burst intervals have been proposed. *Bauer et al., 2017* included a

plasticity parameter that progressively decoupled spatially distributed neural mass units based on the synchrony level. This mechanism accounted for seizure termination, exponential IBI increase and the presence of a transient postictal state. In the model of *Liou et al., 2020* the IBI was constant during seizure expansion and only when the spatial propagation of ictal discharge ceased, the seizure entered the pre-termination stage with a slowing-down trend. An increase in the IBI in this phase was related to the recovery of inhibition and restoration of the Cl⁻ concentration gradient. These studies indicate that spatial properties related to seizure propagation and synchrony are other factors that may affect the evolution of the IBI.

### The postictal period

Seizures are followed by the suppression of physiological rhythms known as postictal EEG suppression (PES) that lasts for seconds or minutes (*Pottkämper et al., 2020*). Using the stimulation protocol, we investigated the duration of PES in the model. Our results showed that shortly after termination of the

**Table 1.** Gating variables of the ionic currents in pyramidal cell model.

| Current | Kinetics/time constant (ms) | |
|---|---|---|
| $I_{Na,soma}$ | $\alpha_m = \frac{0.8(-V-39.8)}{exp\left(\frac{-V-39.8}{4}\right)-1}$ | $\beta_m = \frac{0.7(V+14.8)}{exp\left(\frac{V+14.8}{5}\right)-1}$ |
| | $\alpha_h = 0.32exp\left(\frac{-V-15}{18}\right)$ | $\beta_h = \frac{10}{exp\left(\frac{-V-15}{5}\right)+1}$ |
| $I_{Na,dendrite}$ | $\alpha_m = \frac{0.32(-V-48.9)}{exp\left(\frac{-V-48.9}{4}\right)-1}$ | $\beta_m = \frac{0.28(V+21.9)}{exp\left(\frac{V+21.9}{5}\right)-1}$ |
| | $\alpha_h = 0.128exp\left(\frac{-V-44}{18}\right)$ | $\beta_h = \frac{4}{exp\left(\frac{-V-21}{5}\right)}$ |
| $I_{NaP}$ | $m_\infty = \frac{1}{1+exp\left(\frac{-48.7-V}{4.4}\right)}$ | $\tau_m = \frac{1}{\frac{0.091(V+38)}{1-exp\left(\frac{-V-38}{5}\right)}-\frac{0.062(V+38)}{1-exp\left(\frac{V+38}{5}\right)}}$ |
| | $h_\infty = \frac{1}{1+exp\left(\frac{48.8+V}{9.98}\right)}$ | if $V_m \leq -60$: <br> $\tau_h = 3700 + \frac{2000}{\frac{0.091(V+60)}{1-exp\left(\frac{-V-60}{5}\right)}-\frac{0.062(V+60)}{1-exp\left(\frac{V+60}{5}\right)}}$ <br> if $V_m > -60$: <br> $\tau_h = 1200 + \frac{8000}{\frac{0.091(V+74)}{1-exp\left(\frac{-V-74}{5}\right)}-\frac{0.062(V+74)}{1-exp\left(\frac{V+74}{5}\right)}}$ |
| $I_{Kdr,soma}$ | $n_\infty = \frac{1}{1+exp\left(\frac{-22.8-V}{13.6}\right)}$ | $\tau_m = \frac{1.6(C+exp\left(\frac{V}{5}\right))}{D\exp\left(\frac{-V}{40}\right)(C+exp\left(\frac{V}{5}\right))+0.016exp\left(\frac{V}{5}\right)(64.9+V)}$ <br> $C = -0.00000230599; D = 0.0338338$ |
| $I_{Kdr,dend}$ | $n_\infty = \frac{1}{1+exp\left(\frac{-14.8-V}{13.6}\right)}$ | $\tau_m = \frac{1.6(C+exp\left(\frac{V}{5}\right))}{D\exp\left(\frac{-V}{40}\right)(C+exp\left(\frac{V}{5}\right))+0.016exp\left(\frac{V}{5}\right)(64.9+V)}$ <br> $C = -0.00000230599; D = 0.0338338$ |
| $I_{CaL}$ | $\alpha_m = \frac{1.6}{1+exp(-0.072(V-5))}$ | $\beta_m = \frac{0.02(V+8.9)}{exp\left(\frac{V+8.9}{5}\right)-1}$ |
| $I_{KAHP}$ | $\alpha_m = 2000\left([Ca]_i - [Ca]_{i,rest}\right)$ | $\beta_m = 0.01$ |
| $I_{KC}$, if $V_m \leq -10$ | $\alpha_m = \frac{exp\left(\frac{V+50}{11}-\frac{V+53.5}{27}\right)}{18.975}$ | $\beta_m = 2exp\left(\frac{-V-53.5}{27}\right)$ |
| $I_{KC}$, if $V_m > -10$ | $\alpha_m = 2exp\left(\frac{-V-53.5}{27}\right)$ | $\beta_m = 0$ |
| $I_{KM}$ | $m_\infty = \frac{1}{1+exp\left(\frac{-V+33}{5}\right)}$ | $\tau_m = \frac{1000}{3.3\left[exp\left(\frac{V+35}{40}\right)\right]+\left[exp\left(\frac{-V-35}{20}\right)\right]}$ |

SLE, burst responses were still triggered, however, after few seconds, the excitability decreased and remained reduced for approximately 90 s (*Figure 7B*). The PES duration in our model is consistent with a typical 'seconds to minutes' timescale although available estimates depend on the PES duration assessment method. In single-unit recordings in epileptic patients with neocortical seizures neuronal spiking was fully suppressed for 5–30 s after seizure termination (*Truccolo et al., 2011*). Average duration of postictal suppression based on EEG features in focal seizure patients was estimated at 17 s (*Grigorovsky et al., 2020*; *Table 1*), 120 s (*Payne et al., 2018*; *Table 1*) and around 50–100 s (*Bauer et al., 2017*; *Figure 5*). Our in silico-derived prediction that postictal 'silence' depends on the increased rate of hyperpolarizing $Na^+/K^+$-pump has been previously suggested (*Fisher and Schachter, 2000*) and simulated (*Krishnan et al., 2015*; *Krishnan and Bazhenov, 2011*). In these models, postictal state was generated via both, reduced $[K^+]_o$ to below baseline level and increased $[Na^+]_i$ after termination of an SLE. $[K^+]_o$ decrease below baseline resulted in a negative shift in $E_K$ and membrane hyperpolarization, while elevated $[Na^+]_i$ increased $Na^+/K^+$-pump hyperpolarizing current. A below-reference value of $[K^+]_o$ was indeed observed after seizure termination (*Heinemann et al., 1977*). On the other hand, in other studies, $[K^+]_o$ decayed to baseline level after an SLE offset (*Fisher et al., 1976*; *Futamachi et al., 1974*) and couldn't contribute to the postictal state. Also in our model $[K^+]_o$ undershoot was not observed (*Figure 7C*), suggesting that the main cause of postictal reduction in excitability was hyperpolarizing effect of the $Na^+/K^+$-pump current, which remained elevated above baseline for about 100 s after SLE termination (*Figure 7DE*). The findings generated by our computational model suggest that ion homeostatic processes activated and sustained by the excessive seizure discharges provide a negative feedback mechanism, eventually leading to the cessation of the seizure itself and to the restoration of the normal state after a transitional period of postictal silence.

## Materials and methods

### Geometry

The cell morphology was based on entorhinal cortex PY cells and an interneuron model (*Fransen et al., 2002*), further reduced to equivalent cylinder models. The PY cell consisted of two compartments: a soma with a length of 20 µm and a diameter of 15 µm, and a dendrite with a length of 450 µm and a diameter of 6.88 µm. The interneuron only had a somatic compartment with a length of 20 µm and a diameter of 15 µm. Each compartment was surrounded by its own extracellular space (ECS). The extracellular compartments were embedded in a common bathing medium which represented the surrounding neural tissue and vasculature. The size of the ECS was estimated by the extracellular volume fraction, α defined as the ratio volume of extracellular space/volume of tissue. We used $\alpha$=0.131 which corresponded to the CA1 *st. pyramidale* and a $K^+$ concentration of 3.5 mM (*McBain et al., 1990*).

### Biophysics

The active membrane currents in the PY cell were the fast $Na^+$ and $K^+$ currents ($I_{Na}$ and $I_{Kdr}$, respectively) in both compartments and were responsible for action potential generation; a persistent $Na^+$ current, $I_{NaP}$ in the soma; a high-threshold $Ca^{2+}$ current, $I_{CaL}$ in both compartments; a calcium-dependent afterhyperpolarization $K^+$ current, $I_{KAHP}$ in both compartments; a fast calcium- and voltage-dependent $K^+$ current, $I_{KC}$ in both compartments; and a noninactivating muscarinic $K^+$ current, $I_{KM}$ in the soma. The IN included only the $I_{Na}$ and $I_{Kdr}$ currents responsible for spike generation. All the equations for the active currents were initially based on those described by *Fransen et al., 2002*, however, an additional modification of the parameters described below was required to account for ionic regulation mechanisms. Simulations were performed using the NEURON simulator with a fixed integration step of 0.05ms.

### Passive properties

The reversal potentials were obtained via the Nernst equation:

$$E_X = 2.3 \frac{RT}{zF} log \left( \frac{[X]_o}{[X]_i} \right)$$

**Table 2.** Conductances used in the model.

| Current conductance | Description | Values (S/cm²) |
|---|---|---|
| $g_{Na,leak,PYsoma}$ | $I_{Na,leak}$ conductance in PY soma | $1.5*10^{-5}$ |
| $g_{Na,leak,PYdend}$ | $I_{Na,leak}$ conductance in PY dendrite | $1.1*10^{-5}$ |
| $g_{K,leak,PY}$ | $I_{K,leak}$ conductance in PY soma and dendrite | $3*10^{-5}$ |
| $g_{Cl,leak,PY}$ | $I_{Cl,leak}$ conductance in PY soma and dendrite | $1*10^{-5}$ |
| $g_{Na,PYsoma}$ | $I_{Na}$ conductance in PY soma | 0.014 |
| $g_{Na,PYdend}$ | $I_{Na}$ conductance in PY dendrite | 0.0014 |
| $g_{Kdr,PYsoma}$ | $I_{Kdr}$ conductance in PY soma | 0.032 |
| $g_{Kdr,PYdend}$ | $I_{Kdr}$ conductance in PY dendrite | 0.0032 |
| $g_{NaP}$ | $I_{NaP}$ conductance in PY soma | $60*10^{-5}$ |
| $g_{CaL}$ | $I_{CaL}$ conductance in PY soma and dendrite | $15*10^{-5}$ |
| $g_{KAHP}$ | $I_{KAHP}$ conductance in PY soma and dendrite | $5*10^{-5}$ |
| $g_{KC}$ | $I_{KC}$ conductance in PY soma and dendrite | 0.196*1e3 |
| $g_{KM}$ | $I_{KM}$ conductance in PY soma | 0.006 |
| $g_{Na,leak,IN}$ | $I_{Na,leak}$ conductance in IN | $2.9*10^{-5}$ |
| $g_{K,leak,IN}$ | $I_{K,leak}$ conductance in IN | $6*10^{-5}$ |
| $g_{Cl,leak,IN}$ | $I_{Cl,leak}$ conductance in IN | $1*10^{-5}$ |
| $g_{Na,IN}$ | $I_{Na}$ conductance in IN | 0.013 |
| $g_{Kdr,IN}$ | $I_{Kdr}$ conductance in IN | 0.027 |

where $[X]_i$ and $[X]_o$ are intra- and extracellular concentrations, respectively, of the ions. $X$ = {Na⁺, K⁺, Ca²⁺, Cl⁻, HCO₃⁻}, $F$ is the Faraday constant, $R$ is the gas constant, $z$ is the valence of the ions and $T$=273,16 + 32 is the absolute temperature (*Gnatkovsky et al., 2008*). A leak current, $I_{leak}$, was present in all compartments of both cells and was a sum of the leak currents of Na⁺, K⁺, and Cl⁻, modeled as:

$$I_{i,leak} = g_{i,leak}\left(V - E_i\right)$$

where $g_{i,leak}$ is the leak current conductance of the ion of interest $i$ = {Na⁺, K⁺, Cl⁻}. The resting membrane potential was –61 mV in the pyramidal cell and in the interneuron. The specific axial resistance in both cells was set to $R_a$ = 100 Ohm*cm and the specific membrane capacitance was set to $C_m$ = 1 µF/cm², as in *Fransen et al., 2002*. Based on the $R_a$ and PY cell geometry, the somato-dendritic coupling conductance, $g_c$, was calculated as 1.5 mS.

## Active currents

The original equations used time constant units in seconds (s) and voltage units in volts (V), with 0 V corresponding to the resting membrane potential. All equations were modified to account for the millivolt (mV) and millisecond (ms) units used in our model and the voltage was shifted by –60 mV to correspond to the membrane potential relative to the extracellular space, which was assumed to be 0 mV. Additional modifications of the parameters were required to account for the ionic regulation mechanisms that were not present in the original model. $I_{NaP}$: the activation gate exponent was 2 and the inactivation gate time constant, $\tau_h$, was estimated by fitting the activation function form described by *Fransen et al., 2002* to the experimental data (*Magistretti and Alonso, 1999*). $I_{Kdr}$: the steady-state activation function, $n_{inf}$, and the activation gate time constant, $\tau_n$, were estimated by empirical fit to the experimental data (*Sah et al., 1988*). To increase the firing threshold, the activation curve was shifted toward positive potentials by 18 mV in the soma and 10 mV in the dendrites. The model generated spontaneous fast spiking otherwise. $I_{KAHP}$, $I_{KC}$: these Ca²⁺-dependent currents were modelled according to the model described by *Traub et al., 2003* and were implemented in ModelDB (https://senselab.med.yale.edu/ModelDB/), accession number 20,756. Due to the arbitrary units for

Ca²⁺ concentration in Traub's model, we modified the current formula to correspond to mM units and resting level of [Ca²⁺]ᵢ used in our model. In pyramidal cells, the soma and dendrite membrane potentials, $V_s$ and $V_d$, respectively, were governed by the following Hodgkin-Huxley equations:

$$C\frac{dV_s}{dt} = -I_{Na,soma} - I_{NaP} - I_{Kdr,soma} - I_{CaL} - I_{KAHP} - I_{KC} - I_{KM}$$
$$-I_{leak} - I_{NaKpump} - I_{CaPump} - g_c\left(V_s - V_d\right) - I_{syn}$$

$$C\frac{dV_d}{dt} = -I_{Na,dend} - I_{Kdr,dend} - I_{CaL} - I_{KAHP} - I_{KC} - I_{NaKpump} - I_{CaPump} - g_c\left(V_d - V_s\right) - I_{syn}$$

Transient sodium current

$$I_{Na,soma} = g_{Na,PYsoma}m^3h\left(V - E_{Na}\right)$$
$$I_{Na,dend} = g_{Na,PYdend}m^2h\left(V - E_{Na}\right)$$

Persistent sodium current:

$$I_{NaP} = g_{NaP}m^2h\left(V - E_{Na}\right)$$

Delayed rectifier:

$$I_{Kdr,soma} = g_{Kdr,PYsoma}n^4\left(V - E_K\right)$$
$$I_{Kdr,dend} = g_{Kdr,PYdend}n^2\left(V - E_K\right)$$

High-threshold Ca²⁺ current:

$$I_{CaL} = g_{CaL}m^2\left(V - E_{Ca}\right)$$

Ca²⁺-dependent K⁺ (afterhyperpolarization) current:

$$I_{KAHP} = g_{KAHP}m\left(V - E_K\right)$$

Fast Ca²⁺- and voltage-dependent K⁺ current:

$$I_{KC} = g_{KC}\min([Ca^{2+}]_i/250, 1)m(V - E_K)$$

Muscarinic current:

$$I_{KM} = g_{KM}m\left(V - E_K\right)$$

Equations of gating variables are given in *Table 1*. Conductance values are given in *Table 2*. Membrane potential of the interneuron was governed by the following Hodgkin-Huxley equations:

$$C\frac{dV}{dt} = -I_{Na} - I_{Kdr} - I_{leak} - I_{NaKpump} - I_{syn} - I_{stim}$$

Current equations, kinetics and time constants of these currents were the same as in pyramidal cell soma. To prevent depolarization block of the IN during current stimulation, activation curve of $I_{Na}$ was shifted 3 mV toward more negative potential values, while activation curve of $I_{Kdr}$ was shifted by 19 mV toward more negative potential values.

## Ionic dynamics

The model included six types of ions (K⁺, Na⁺, Cl⁻, Ca²⁺, A⁻, and HCO₃⁻) with variable intra- and extracellular concentrations, except for HCO₃⁻, for which equilibrium is rapidly attained (*Theparambil et al., 2020*). The evolution of the ion concentrations was based on the following equations:

$$\frac{d[K^+]_i}{dt} = J_K^i + J_{K,longitudinal}^i + J_{K,KCC2}^i + J_{K,Pump}^i + J_{K,vol}^i$$

$$\frac{d[K^+]_o}{dt} = J_K^o + J_{K,radial} + J_{K,longitudinal}^o + J_{K,bath} + J_{K,KCC2}^o + J_{K,Pump}^o + J_{glia} + J_{K,vol}^o$$

$$\frac{d[Na^+]_i}{dt} = J_{Na}^i + J_{Na,longitudinal}^i + J_{Na,Pump}^i + J_{vol}^i$$

$$\frac{d[Na^+]_o}{dt} = J^o_{Na} + J_{Na,radial} + J^o_{Na,longitudinal} + J_{Na,bath} + J^i_{Na,Pump} + J^o_{Na,vol}$$

$$\frac{d[Cl^-]_i}{dt} = J^i_{Cl} + J^i_{Cl,GABAa} + J^i_{Cl,longitudinal} + J^i_{Cl,KCC2} + J^i_{Cl,vol}$$

$$\frac{d[Cl^-]_o}{dt} = J^o_{Cl} + J^o_{Cl,GABAa} + J^o_{Cl,longitudinal} + J_{Cl,bath} + J^o_{Cl,KCC2} + J^o_{Cl,vol}$$

$$\frac{d[Ca^{2+}]_{i,tot}}{dt} = J^i_{Ca} + J^i_{Ca,longitudinal} + J^i_{CaPump} + J^i_{Ca,vol}$$

$$\frac{d[Ca^+]_o}{dt} = J^o_{Ca} + J^o_{Ca,longitudinal} + J^o_{CaPump} + J^o_{Ca,vol}$$

$$\frac{d[A^-]_i}{dt} = J^i_{A,vol}$$

where $[Ca^{2+}]_{i,tot}$ is the total intracellular calcium concentration (see calcium buffer below). All fluxes (mM/ms) are specified below.

## Membrane currents

The contribution of transmembrane currents to variations in intra- and extracellular ion concentrations was obtained via the following equations:

$$J^i_X = \frac{-\sum I_X S}{zFV_i}$$

$$J^o_X = \frac{\sum I_X S}{zFV_o}$$

where the sum of $I_X$ is a net membrane current carrying ion X, S is the surface area of the compartment, z is the valence of the ions, F is the Faraday constant and $V_i$ and $V_o$ are the volumes of the intra- and extracellular compartments.

## Longitudinal diffusion

Longitudinal diffusion of $K^+$, $Na^+$, $Ca^{2+}$ and $Cl^-$ was implemented between the somatic and dendritic compartments in the intracellular and extracellular space of the same cell. It was described by Fick's first law:

$$J^{io}_{X,longitudinal} = D_x \frac{\left([X]_{io,a} - [X]_{io}\right)S}{LV_{io}}$$

where $D_x$ is the diffusion coefficient for the ion X, $[X]_{io}$ is the ion concentration in a given intra- or extracellular compartment, $[X]_{io,a}$ is the ion concentration in the adjacent compartment, S is the cross-sectional area between the compartments, $V_{io}$ is the compartment volume, L is the distance between the centers of the compartments. The diffusion coefficients were (in um²/ms): $D_{Na}$ = 1.33, $D_K$ = 1.96, $D_{Ca}$ = 0.6, $D_{Cl}$ = 2.03 (as in **Somjen et al., 2008**).

## Radial diffusion

$Na^+$ and $K^+$ ions diffused radially between adjacent extracellular compartments modeled as concentric shells around the neurons. The radial exchange of ions between adjacent shells was described by Fick's first law:

$$J_{X,radial} = D_x \frac{\sum\left([X]_{o,a} - [X]_o\right)S}{drV}$$

where $[X]_o$ is the ion concentration in a given shell and the sum goes over all adjacent shells having concentrations $[X]_{o,a}$, V is a given shell volume, dr is the distance between the centers of the shells and S is the surface contact area between the shells calculated at 16% of the total outer shell surface. The electrostatic drift of ions was neglected as the ion movement due to the electrical potential gradient in the extracellular space was small compared to the diffusion.

## Diffusion to/from the bath

Radial diffusion of $K^+$, $Na^+$ and $Cl^-$ between the ECS and the bath was described by Fick's first law:

$$J_{X,bath} = \frac{1}{s} D_x \frac{\left([X]_{bath} - [X]_o\right)S}{drV}$$

where $D_x$ is the diffusion coefficient for the ion $X$, $s$ is the scaling constant, $S$ is the outer surface of the shell, $[X]_{bath}$ is the bath concentration of the ion $X$, $[X]_o$ is the ion concentration in a given shell, $V$ is the shell volume, $dr$ is the distance between the extracellular space and the bath (assumed to be half of the shell thickness). Flux $J_{bath}$ represents various processes such as diffusion to more distant areas of the brain and cerebrospinal fluid, active transport of potassium into capillaries and potassium spatial buffering by astrocytes. The effective time constant of these joint processes is likely to be much slower than that of radial and longitudinal diffusion and is described by the scaling constant $s=4.4*10^4$.

## The Na$^+$/K$^+$ pump

The Na$^+$/K$^+$ pump was modeled as the sodium and potassium transmembrane currents (**Kager et al., 2000**):

$$I_{Na,Pump} = 3I_{max} flux([Na^+]_i, [K^+]_o)$$

$$I_{K,Pump} = -2I_{max} flux([Na^+]_i, [K^+]_o)$$

$$flux([Na^+]_i, [K^+]_o) = \left(1 + \frac{Km_K}{[K^+]_o}\right)^{-2} \left(1 + \frac{Km_{Na}}{[Na^+]_i}\right)^{-3}$$

with $Km_K$ = 2 mM and $Km_{Na}$ = 10 mM. $I_{max}$ values were computed for each cell and compartment to balance Na$^+$ and K$^+$ membrane currents at rest and were as follows (in mA/cm$^2$): 0.014 (PY soma), 0.009 (PY dendrite), 0.025 (IN).

## KCC2 cotransport

The KCC2 cotransporter currents were modeled according to **Wei et al., 2014a**:

$$I_{K,KCC2} = U_{KCC2}\, log\left(\frac{[K^+]_i[Cl^-]_i}{[K^+]_o[Cl^-]_o}\right)$$

$$I_{Cl,KCC2} = -I_{K,KCC2}$$

with cotransporter strength adjusted to balance chloride leak current at rest, $U_{KCC2}$=0.002 mA/cm$^2$.

## Glial uptake

Potassium uptake by the glia was modeled as a set of differential equations (**Kager et al., 2000**):

$$J_{glia} = -k_2[K^+]_o[B] + k_1[KB]$$

$$\frac{d[B]}{dt} = -k_2[K^+]_o[B] + k_1[KB]$$

$$\frac{d[KB]}{dt} = k_2[K^+]_o[B] - k_1[KB]$$

where $[B]$ is the free buffer, $[KB]$ is the bound buffer (= $[B]_{max}$–$[B]$), $B_{max}$ = 1100 mM. $k_1$=0.0008 ms$^{-1}$ and $k_2 = \frac{k_1}{1+exp\left(\frac{[K]_o-16}{-1.25}\right)}$ are backward and forward rate constants, respectively.

## The calcium pump and buffer

The calcium pump and buffer which altered the intracellular Ca$^{2+}$ were modeled according to the model implementation of **Somjen et al., 2008** in ModelDB, accession number 113,446. The calcium pump which extruded Ca$^{2+}$ from the cells was modeled as a Ca$^{2+}$ transmembrane current:

$$I_{CaPump} = \frac{I_{max}}{1+\frac{K_{pump}}{[Ca^{2+}]_i}}$$

with $I_{max}$ = 2.55 mA/cm$^2$ and $K_{pump}$ = 0.0069 mM. Intracellular Ca$^{2+}$ was buffered by first-order chemical Ca$^{2+}$ buffer with a total concentration of $[B]_i$ and an equilibrium constant of $K_d$. Calcium buffering was fast and under the assumption of equilibrium conditions, the relationship between the total and free intracellular calcium concentrations, $[Ca^{2+}]_{i,tot}$ and $[Ca^{2+}]_i$, was given by **Borgdorff, 2002**, pg. 27:

$$[Ca^{2+}]_{i,tot} = [Ca^{2+}]_i \frac{[B]_i + K_d + [Ca^{2+}]_i}{K_d + [Ca^{2+}]_i}$$

where $[B]_i$ = 1.562 mM*$(V_i^0/V_i)$, $V_i$ is the intracellular compartment volume, $V_i^0$ is the intracellular compartment volume at rest and $K_d$ = 0.008 mM.

## Volume changes

Volume changes were modeled according to *Somjen et al., 2008*. The rates of intra- and extracellular volume changes were proportional to the difference in osmotic pressure between the intra- and extracellular compartments and fulfilled the conservation of total volume.

$$\frac{dV_i}{dt} = \Delta$$

$$\frac{dV_o}{dt} = -\Delta$$

where

$$\Delta = \frac{c(\pi_i - \pi_o)}{\tau}$$

$$\pi_i = [Na^+]_i + [K^+]_i + [Cl^-]_i + [Ca^{2+}]_i + [HCO_3^-]_i + [A^-]_i$$

$$\pi_o = [Na^+]_o + [K^+]_o + [Cl^-]_o + [Ca^{2+}]_o + [HCO_3^-]_o + [A^-]_o$$

$V_i$ and $V_o$ are the volumes of the intra- and extracellular compartments and $\tau$=250ms. The constant c is introduced for unit conversion and is equal to 1 µm³/mM, hence units of $\Delta$ are µm³/ms. The extracellular volume was initially 15% of the cellular volume and was allowed to shrink maximally down to 4%. Volume changes affected concentrations but not the total mass of each ion within a compartment. The conservation of mass required additional fluxes:

Intracellular

$$J_{X,vol}^i = \frac{-\Delta}{V_i}\left[X\right]_i$$

Extracellular

$$J_{X,vol}^o = \frac{\Delta}{V_o}\left[X\right]_o$$

Extracellular space (ES) volume shrinks maximally by about 27%, comparable to average reduction of 30% during self-sustained epileptiform discharges (*Dietzel et al., 1980*). Intracellular space (IS) volume expands about 4%, being a consequence of constant total volume (ES +IS = const). Volume changes and resulting shifts in representative ion concentrations are shown in *Appendix 1—figure 4*. We note that volume changes in our model didn't have big impact on the observed model dynamics. Shifts in ionic gradients contributed by slow volume changes were efficiently compensated by other homeostatic mechanisms, which acted on a faster time scale. In the real tissue astrocyte swelling may be significant and may reduce flow of ions and oxygen (affecting Na⁺/K⁺-pump activity) contributing to seizures and spreading depression (*Hübel and Ullah, 2016*). In our model, glial cell swelling was not included and its effects on diffusion and the Na⁺/K⁺-pump were not simulated which is one of the model limitations.

## Initial ion concentrations

The initial concentrations were based on existing literature: $[Na^+]_i$, $[Ca^{2+}]_i$, $[Na^+]_o$, $[K^+]_o$, $[Ca^{2+}]_o$ and $[A^-]_o$ (*Somjen et al., 2008*); $[Cl^-]_i$, $[HCO_3^-]_i$, $[HCO_3^-]_o$ (*Doyon et al., 2011*); $[K^+]_i$ $[Cl^-]_o$ (*Payne et al., 2003*). In setting chloride and potassium concentrations, we additionally aimed to fulfill $E_K < E_{Cl}$ and $E_{GABAa} \sim$ –70 mV (*Andersen et al., 1980*). The $[A^-]_i$ concentration was set to fulfill osmotic equilibrium condition. Hence, the initial concentrations were as follows (in mM): $[Na^+]_i$ = 10, $[Na^+]_o$ = 140, $[K^+]_i$ = 87, $[K^+]_o$ = 3.5, $[Cl^-]_o$ = 135, $[Cl^-]_i$ = 6, $[Ca^{2+}]_i$ = 5e-5, $[Ca^{2+}]_o$ = 2, $[A^-]_i$ = 187.5, $[A^-]_o$ = 0, $[HCO_3^-]_i$ = 15, $[HCO_3^-]_o$ = 25. These values gave the following Nernst potentials (in mV): $E_{Na}$ = 69.4, $E_K$ = –84.5, $E_{Cl}$ = –81.8, $E_{Ca}$ = 139.3, $E_{HCO3}$ = –13.4, $E_{GABAa}$ = –69.5.

## Resting state

Steady-state conditions at rest were characterized by no net flux of the ions at the resting potential (–61 mV). These conditions were determined separately for each cell and compartment. For chloride, KCC2 cotransport strength $U_{KCC2}$ was adjusted to balance Cl⁻ leak current at rest, that is:

$$I_{Cl,leak} = -I_{Cl,KCC2}$$

For sodium and potassium, first, Na$^+$ leak current conductance $g_{Na,leak}$ was adjusted to ensure that at rest all passive and voltage-gated membrane currents $I_{Na}$ and $I_K$ are in the ratio –3/2, that is,:

$$I_{Na} = \frac{-3}{2} I_K$$

Next, the Na$^+$/K$^+$-pump strength $I_{max}$ was adjusted such that $I_{Na}$ and $I_K$ currents were balanced by equal and opposite pump currents:

$$-I_{Na} = I_{Na,pump}$$
$$-I_K = I_{K,Pump}$$

## Synaptic connections and model inputs

The pyramidal cells created excitatory AMPA synaptic connections with the interneuron and all other pyramidal cells. The interneuron created an inhibitory GABAa synaptic connection with each pyramidal cell. Excitatory synapses were placed in the middle of the PY dendrite and the middle of the IN soma. Inhibitory synapses were placed in the middle of the PY soma. The time course of synaptic conductance was modeled with a built-in NEURON mechanism Exp2Syn, implementing a dual exponential function:

$$g = g_{max} \left[ exp \left( \frac{-t}{\tau_2} \right) - exp \left( \frac{-t}{\tau_1} \right) \right]$$

where the rise and decay time constants, $\tau_1$ and $\tau_2$, were 2ms and 6ms, respectively, for all synapses. $g_{max}$ = weight*factor, where the factor was defined so that the normalized peak was 1. The weights for the synapses between the PY and from the PY to the IN were $w_{ee}$ = 0.0002 μS and $w_{ei}$ = 0.0017 μS, respectively. The inhibitory synaptic weight, $w_{ie}$, was 0.0005 μS. All pyramidal cells received background input modeled as a Poisson spike train, which was different in each cell, activated an excitatory synapse at a rate of 5 Hz and had a synaptic weight $w_{input}$ = 0.0004 μS. To initiate the SLE, a depolarizing ramp current, $I_{inj}$, was injected into the interneuron, with the initial amplitude of 0.35 nA linearly decreasing toward 0 over 40 s.

The inhibitory GABA$_a$ postsynaptic currents were carried by Cl$^-$ and HCO$_3^-$ ions (**Jedlicka et al., 2010**):

$$I_{GABAa} = I_{ClGABAa} + I_{HCO3GABAa}$$
$$I_{ClGABAa} = (1 - P) g (V - E_{Cl})$$
$$I_{HCO3GABAa} = P g (V - E_{HCO3})$$
$$E_{GABAa} = (1 - P) E_{Cl} + P E_{HCO3}$$

where relative permeability $P$ was 0.18.

## Calculation of the LFP

The local field potentials were calculated based on all transmembrane currents in all cells using the following equation (**Nunez and Srinivasan, 2006**):

$$\phi (r, t) = \frac{1}{4 \pi \sigma} \sum_{n=1}^{N} \frac{I_n (t)}{|r - r_n|}$$

where $I_n$ is a point current source at position $r_n$ (taken as the position of a mid-point of a compartment) and $r$ is the position of the electrode. $\sigma$=0.3 S/m is the extracellular conductivity (**Lindén et al., 2014**). The electrode was located in the middle of the somatic layer of the PY and IN cells, approximately 16 μm from the centers of the somas of two neighbouring PY cells. The currents from the interneuron were taken with a weight of 0.2 to decrease their contribution. Also, the influence of the injected current on the LFP was removed. The amplitude of the simulated LFP signal was an order of magnitude smaller than the experimental data. This was due to current point-source approximation and the small number of cells in the modeled network.

## Inter-burst interval fitting

Inter-burst intervals were fitted in Matlab with linear, exponential, logarithmic and square root relationships. The linear function, $IBI(t)=A + Bt$, was fitted using the *polyfit* procedure. The exponential function, $IBI(t)=A + Bexp(Ct)$, was fitted using the *fminsearch* procedure. The logarithmic and square root fits were based on the bifurcation theory, which suggests that close to bifurcation, the oscillation frequency may be constant or decay as square root or inverse of a logarithm of the distance to the bifurcation (*Izhikevich, 2000*). Accordingly, we fitted IBI (i.e. inverse of frequency) with logarithmic and inverse square root functions $IBI(\lambda)=A + Blog(\lambda)$, where $log()$ denotes the natural logarithm function, and $IBI(\lambda)=A + B/sqrt(\lambda)$, where $sqrt()$ denotes the square root function. Both functions were fitted using the *polyfit* procedure with $log(\lambda)$ and $1/sqrt(\lambda)$ treated as a predictor variable. The distance to the bifurcation point was computed as $\lambda=t_{end} - t+1$, where $t_{end}$ denotes bifurcation point, that is, time of an SLE end while $t$ is time since the beginning of an SLE. One second offset in $\lambda$ was necessary to avoid infinity in the predictor variables when $t=t_{end}$. The goodness of fit was evaluated by Root Mean Square Error (RMSE).

## Software accessibility

The model is publicly available in the ModelDB (http://modeldb.yale.edu/267499).

## Acknowledgements

We are grateful to Laura Uva and Laura Librizzi for helpful discussions throughout the preparation of this work and for providing the SLE data from the in vitro isolated whole guinea pig brain. We thank Laura Tassi from the *Claudio Munari* Epilepsy Surgery Center of the Niguarda Hospital in Milano, Italy, for providing the intracerebral data on human seizures. The work of MdC and VG was supported by the EPICARE grant of the Associazione Paolo Zorzi for the Neuroscience.

## Additional information

### Funding

| Funder | Grant reference number | Author |
| --- | --- | --- |
| Associazione Paolo Zorzi for the Neuroscience | EPICARE | Marco de Curtis Vadym Gnatkovsky |

The funders had no role in study design, data collection and interpretation, or the decision to submit the work for publication.

### Author contributions

Damiano Gentiletti, Software, Investigation, Visualization, Methodology; Marco de Curtis, Conceptualization, Resources, Data curation, Validation, Writing – original draft, Writing – review and editing; Vadym Gnatkovsky, Resources, Data curation, Validation, Writing – review and editing; Piotr Suffczynski, Conceptualization, Formal analysis, Supervision, Validation, Investigation, Visualization, Methodology, Writing – original draft, Writing – review and editing

### Author ORCIDs

Vadym Gnatkovsky  http://orcid.org/0000-0002-4543-0464
Piotr Suffczynski  http://orcid.org/0000-0002-2300-1415

### Decision letter and Author response

Decision letter https://doi.org/10.7554/eLife.68541.sa1
Author response https://doi.org/10.7554/eLife.68541.sa2

## Additional files

### Supplementary files
• Transparent reporting form

## Data availability

All experimental data and analysis code is provided. Source data and analysis code (Matlab) is provided for Figure 2. Source data is provided for Figure 3A Source data and analysis code (Matlab) is provided separately for Figure 6A, 6B, 6C. Source data and analysis code (Matlab) is provided for Figure 6 - figure supplement 1. The model NEURON files with a code reproducing Figure 2 (main simulation results) is publicly available at Model DB database (http://modeldb.yale.edu/267499).

The following dataset was generated:

| Author(s) | Year | Dataset title | Dataset URL | Database and Identifier |
|---|---|---|---|---|
| Suffczynski P, Gentiletti D | 2022 | A focal seizure model with ion concentration changes | http://modeldb.yale.edu/267499 | ModelDB, 267499 |

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

## Appendix I

**A**

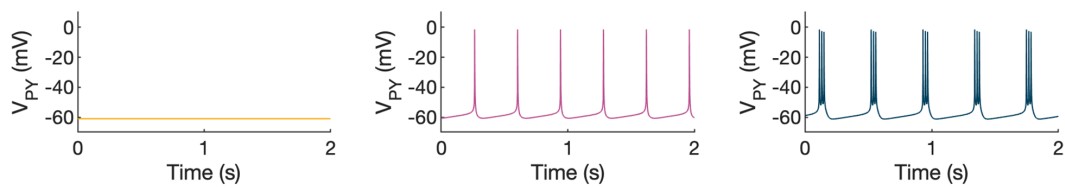

**B**

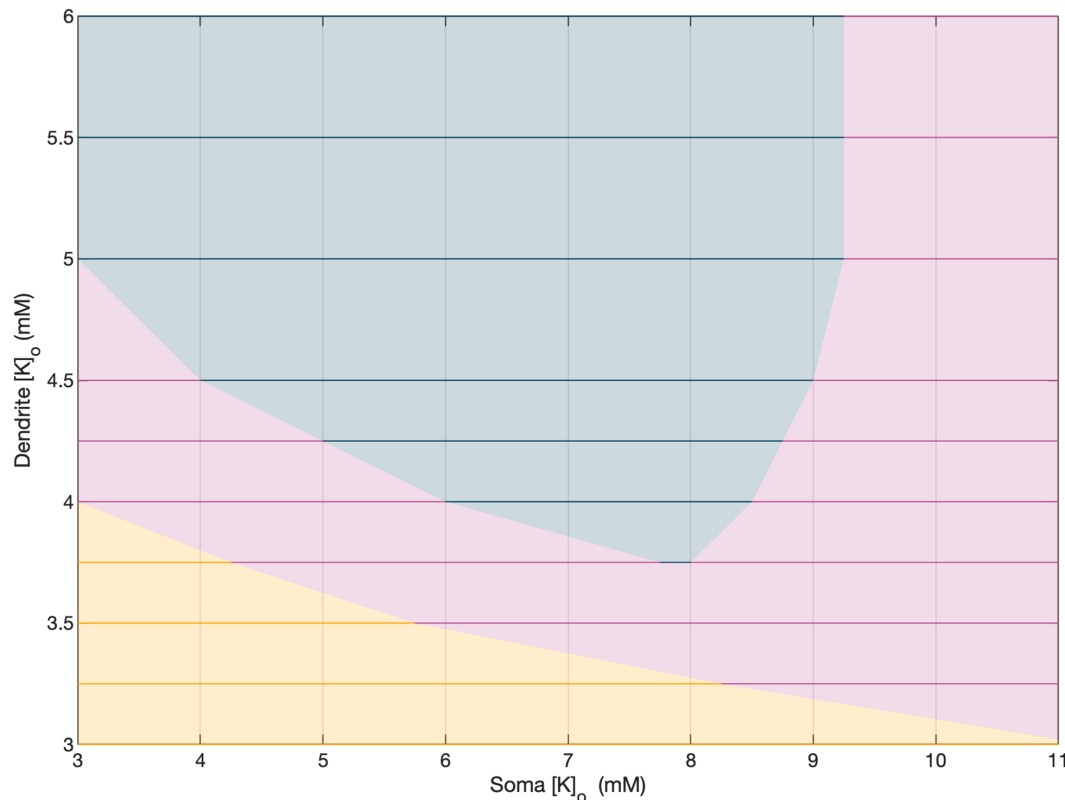

**Appendix 1—figure 1.** A bifurcation diagram of a single PY cell. The 2D bifurcation diagram demonstrates the behavior of a single PY cell as a function of extracellular potassium concentration in the dendritic ($[K^+]_{o,dend}$) and somatic compartments ($[K^+]_{o,soma}$) used as control parameters. Concentrations of all other ions were fixed at their reference values (except chloride: $[Cl^-]_{i,soma}$, $[Cl^-]_{i,dend}$ equal to 7 mM), all ion accumulation mechanisms were blocked and all synaptic connections were removed. (**A**) The three graphs show the PY cell activity traces for different values of the control parameters: resting (yellow, $[K^+]_{o,soma}$ = 3.5 mM, $[K^+]_{o,dend}$ = 3.5 mM), tonic firing (violet, $[K^+]_{o,soma}$ = 4.5 mM, $[K^+]_{o,dend}$ = 4 mM) and bursting (dark blue, $[K^+]_{o,soma}$ = 6.5 mM, $[K^+]_{o,dend}$ = 4 mM). In each panel, 2 seconds of activity is shown. (**B**) The colors of the 2D diagram correspond to the types of activity shown above in (**A**). For low $[K^+]_{o,dend}$ and $[K^+]_{o,soma}$ the cell was at rest. A moderate increase in either $[K^+]_{o,dend}$ or $[K^+]_{o,soma}$, or both, led to tonic firing. Subsequent increases in these parameters led to bursting.

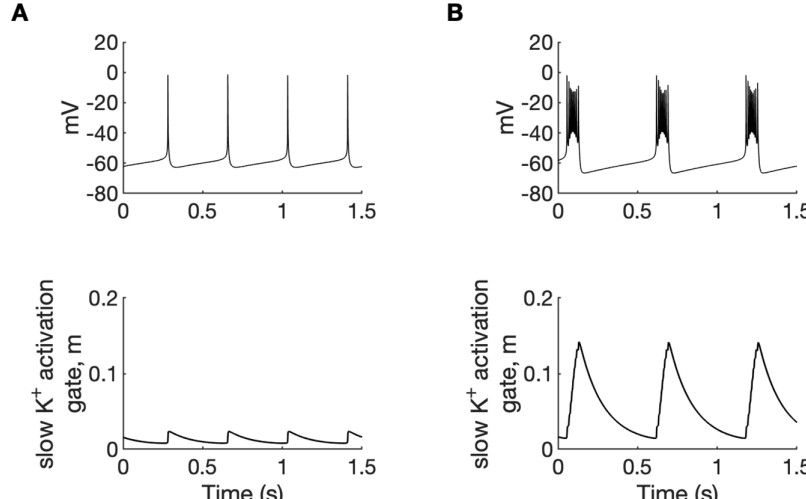

**Appendix 1—figure 2.** A comparison of tonic firing (**A**) and bursting (**B**) of a PY cell. In each column, the membrane potential of a cell (top) and the activation gate of $I_{KM}$ (bottom) is shown. During tonic firing, the activation gate of the M-type potassium current was closed. The prolonged depolarization of the cell during bursting led to the opening of the activation gate $m$ and activation of the $I_{KM}$ current, which eventually terminated the burst. The simulation was performed using an isolated PY cell model with the concentrations of all ions fixed at their reference values, except $[Cl^-]_{i,soma}$ = 7, $[Cl^-]_{i,dend}$ = 7 mM and (**A**): $[K^+]_{o,soma}$ = 3.5 mM, $[K^+]_{o,dend}$ = 4.5 mM, (**B**): $[K^+]_{o,soma}$ = 5.25 mM and $[K^+]_{o,dend}$ = 4.5 mM.

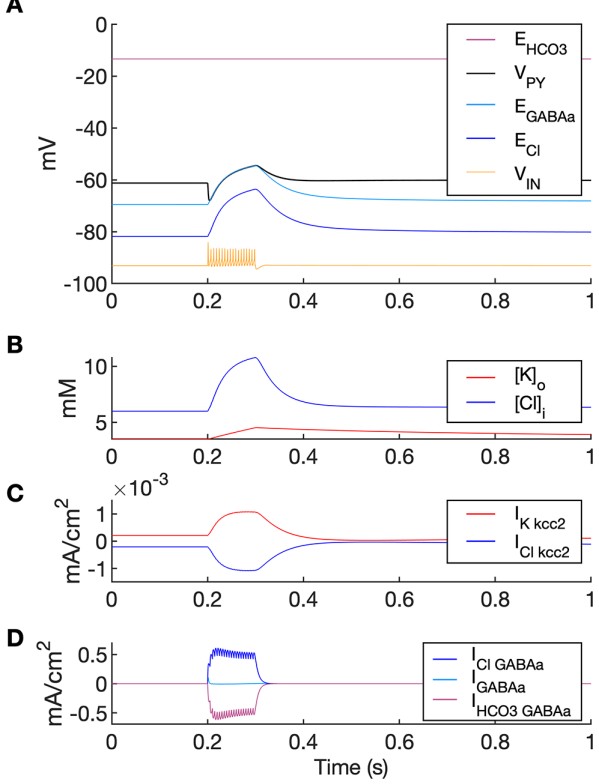

**Appendix 1—figure 3.** A simulation of biphasic GABAa response. In this simulation size of both PY compartments was scaled down by factor 10, to represent small dendritic compartments. Single GABAa synapse was located in a segment having diameter 1.5 um and length 2 um (i.e., having volume 1,000 times smaller and GABAa conductance density 100 times larger than soma in the original model). (**A**) High frequency IN firing (shown schematically by $V_{IN}$, yellow; not to scale) was induced by IN current stimulation of 100ms duration. GABAa

*Appendix 1—figure 3 continued*
receptor-mediated postsynaptic potential response consisted of the initial hyperpolarization followed by a long-lasting depolarization ($V_{PY}$, black). Chloride accumulation ((**B**), blue) was mediated by large Cl⁻ influx via GABAa receptor as compared to Cl⁻ extrusion via KCC2 ((**C**) vs. (**D**)). Accordingly, the biphasic potential resulted from positive shift in $E_{Cl}$ ((**A**), blue) and relatively high $E_{HCO3}$ ((**A**), violet) leading to depolarizing shift in $E_{GABAa}$ ((**A**), light blue). During GABAa receptor activation $V_{PY}$ was clamped to $E_{GABAa}$ due to large GABAa conductance density.

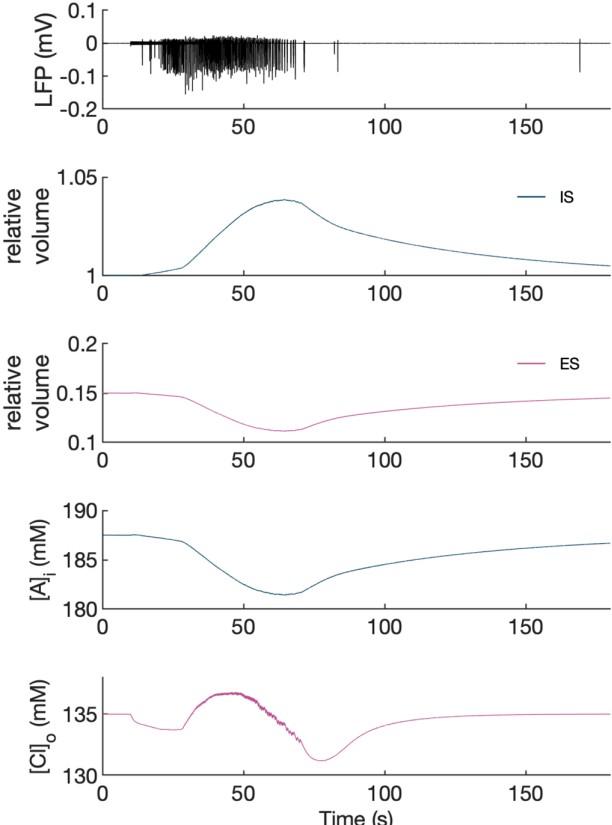

**Appendix 1—figure 4.** Volume changes during an SLE in the model. The panels show from top to bottom: LFP, relative volume changes and representative changes in intracellular A⁻ ion and extracellular Cl⁻ concentration in the PY somatic compartment during an SLE. [A⁻]ᵢ was affected only by volume changes while [Cl⁻]ₒ was additionally affected by inward chloride leak and GABAa currents, KCC2 and Cl⁻ diffusion to the bath. It can be seen that an increase in intracellular space (IS) volume (second panel, dark blue) is exactly mirrored by a decrease in [A⁻]ᵢ. A decrease in extracellular space (ES) volume (third panel, violet) gives rise to an increase in [Cl⁻]ₒ above baseline despite Cl⁻ influx into the cells, in agreement with the experimental data (***Dietzel et al., 1982***).

