## [Editor Report]

In this manuscript the authors build a small-scale biophysically realistic network model to study seizure dynamics which incorporates Hodgkin Huxley mechanisms and ion dynamics. The model enhances our understanding of the mechanisms underlying the evolution and termination of focal seizures. In particular it demonstrates that intense activation of inhibitory interneurons, by driving changes in transmembrane ion dynamics are a possible mechanism for driving the initiation and prolongation of seizures.

---

## [Decision Letter]

**Decision letter after peer review:**

[Editors’ note: the authors submitted for reconsideration following the decision after peer review. What follows is the decision letter after the first round of review.]

Thank you for submitting your work entitled "Focal seizures are organized by feedback between neural activity and ion concentration changes" for consideration by *eLife*. Your article has been reviewed by 3 peer reviewers, and the evaluation has been overseen by a Reviewing Editor and a Senior Editor. The following individuals involved in review of your submission have agreed to reveal their identity: Joseph V Raimondo (Reviewer #1); Maxim Bazhenov (Reviewer #2).

Comments to the Authors:

We are sorry to say that, after consultation with the reviewers, we have decided that your work will not be considered further for publication by *eLife*.

Although there was certainly enthusiasm about the model, which was found to be generally thorough and broadly satisfying in its behaviour and predictions, there was ultimately collective concern from all 3 reviewers about whether there was the requisite level of novelty and advance over prior work to justify publication in *eLife*.

Please find the reviews below.

*Reviewer #1 (Recommendations for the authors):*

The model uses the NEURON framework and is well put together. On the whole its behaviour is both satisfying and reassuring. The model recapitulates the electrographic behaviour of animal model and human EEG recordings in particular the low-voltage fast activity preceding the tonic phase of a seizure, which transitions into bursting. The bursts then slow in frequency before postictal suppression of activity is observed. These dynamics emerge out of the model due to its inclusion of ion dynamics of K^+^, Na^+^ and Cl^-^ and multiple cellular mechanisms including accounting for excitatory and inhibitory cell populations, dendritic and somatic compartments, diffusion, glial buffering, ion channels and ion transporters.

Whilst the work is very much worthy of publication, I am not convinced that it generates sufficiently novel findings and advances over previous work in the field (e.g. Krishnan 2011 and Krishnan 2015 and others) to be of sufficient interest to warrant publication in *eLife*. In my opinion the importance of K^+^, Cl^-^ and Na^+^ dynamics for seizure evolution have been demonstrated before. (e.g. increases in K^+^ driving SLE initiation, transition between tonic and bursting activity and seizure cessation occurring due to enhanced Na+/K^+^ ATPase activity).

(1) I am concerned as to the applicability of the experimental data to this model. SLEs in the whole guinea pig brain were elicited using bicuculline, which blocks GABAaR transmission yet current from the IN to PYs via activated GABAaRs is presumably an important component of the computational model?

(2) Understandably the model cannot recapitulate all biological detail but there are some aspects about the model which seem odd and would need justification:

a. e.g. Intracellular HCO_3_^-^ is higher than intracellular bicarbonate (15 mM) being higher than extracellular bicarbonate (11 mM). This suggests that intracellular pH is more alkaline than extracellular pH which is not the case. In addition, this would make the bicarb reversal +8 mV which is high and unrealistic.

b. In the neurons the Na^+^ leak conductance is 40% of the K^+^ leak conductance, this seems very high and not a typical ratio of permeabilities for neurons.

c. It wasn't clear whether the leak conductances for the various ions were actually contributing to the ion dynamics, e.g. was Cl^-^ flux through the baseline Cl^-^ leak conductance in neurons contributing to changes in [Cl^-^]i?

d. Ek = Ecl at rest, this is also not physiological, I understand why this was done so that KCC2 was at equilibrium at baseline, but this is not ideal. Rather there should be a tonic Cl^-^ leak influx ensuring that "at baseline" Ecl>Ek as observed experimentally.

(3) Modelling of Cl^-^ changes especially Cl^-^ flux through GABAaRs. I am also not entirely sure that Cl^-^ flux through GABAaRs was modelled correctly to capture the potential for biphasic/ depolarizing responses via intensely activated GABAaRs (the authors should note amongst others Ruusuvuori 2004). The calculation of Egaba included HCO_3_^-^ (line 833) but how was Icl (Cl^-^ flux) via activated GABAaRs calculated? Ie in the model if Vm is at the GABAaR reversal potential (Egaba) would Cl^-^ ion flux into the cell via GABAaRs (Icl) be zero? Ie in the model if Icl calculated as 4/5 of Igaba and Igaba = Ggaba(Vm – Egaba), then when Vm = Egaba, Igaba and consequently Icl is 0. This shouldn't be the case. Rather Igaba = Icl + Ihco3 where Icl = gcl(Vm – Ecl) and Ihco3 = ghco3(Vm – Ehco3) and ggaba = 4/5xgcl +1/5xghco3. Seeing as Cl^-^ accumulation is a fundamental part of the model this should have been made more clear. E.g. in Figure 5, could Icl via GABAaRs also be plotted? I worry as the Cl^-^ influx in the model seemed to be coming predominantly via KCC2 (due to the raised extracellular K^+^) whereas experimentally this is likely also coming predominantly through GABAaRs. This is reflected in the time course of [Cl^-^]i changes in the model which are slower than [K^+^]o and continue increasing until the end of the SLE which is not typically what is observed experimentally (see intracellular Cl^-^ recordings e.g. Raimondo 2013).

(4) Perhaps the authors could make it more clear that the exact same experimental data was also presented in Gentilleti 2017.

*Reviewer #2 (Recommendations for the authors):*

Gentiletti et al. investigated potential role of non-synaptic mechanisms driving seizure-like event (SLE) generation. Using a detailed computational model of a small network of neurons, the authors demonstrate that the complex interaction between specific ion species may give rise to SLE. The detailed analysis of the computational model provides an interesting approach to developing a unifying framework for neural dynamics. The strength of this manuscript is in the direct validation of key aspects of the computational model using in vitro electrophysiology. Additionally, predictions made by the model such as the slowing of inter-burst intervals are subsequently validated in both human and mouse data. The conclusions made by the authors of this manuscript are supported by their results and are in line with previous work in the field. Finally, the manuscript does a good job relating the novel results of this new manuscript with established results in the field.

1. A strength of this manuscript is the direct validation of the computational model with experimental results. Given the multitude of methods available for inducing seizure-like events (SLE) in vitro, it is a bit surprising that the authors chose to use an arterial application of bicuculline (Figure 3). Bicuculline is a competitive antagonist of the GABA-A receptor resulting in the reduction of inhibitory GABAergic signaling. However, the SLE induced in the model is caused by an increase in inhibitory activity, through direct depolarization, rather than a decrease in GABAergic signaling. At first glance these methods for inducing SLE seem to be at odds with one another. Given the observed increase in extracellular K^+^ at SLE onset, this mismatch in the method for SLE generation in vitro and in silico may further highlight one of the primary claims of the manuscript specifically that disruption of ionic homeostasis rather than solely synaptic excitatory/inhibitory imbalance is a mechanism for SLE generation. Additional discussion of this observation and similar phenomena in other methods for SLE generation in vitro would further strengthen this interesting point.

2. Bifurcation analysis in figure 4A produces interesting results that are in line with and supported by previous work. As stated in the text, an assumption made by the authors is that the intracellular dendritic and somatic Na^+^ is equal. The authors further mention that this assumption may result in an overestimation of dendritic Na^+^ and Na/K pump activity but do not discuss how this might impact the results presented in the bifurcation diagrams. Please include.

3. It is striking that the time course of the extracellular dendritic K^+^ concentration is much slower than for the soma. It is not clear if the delayed increase in dendritic K^+^ is a prediction of the computational model or if it has been experimentally observed and incorporated into the model as such. Some discussion clarifying this point is needed. To that point, are the authors suggesting that the longitudinal diffusion of K^+^ from soma to dendrite is driving the delayed increase? If so, how might the observed dynamics change if the direction was reversed? The computational model contains an inhibitory neuron which seems to target specifically the soma. For this reason, it is not surprising that the somatic K^+^ increases first. However, peri-somatic inhibition is a characteristic of PV-inhibitory interneurons. Somatostatin (SOM) expressing inhibitory interneurons predominantly target dendrites. Given recent studies showing that stimulation of either SOM or PV interneurons can trigger seizure onset, how might this impact the bifurcation dynamics presented here?

4. The results pertaining to Cl concentration are interesting and extend a large number of recent studies examining the role of Cl in seizure dynamics. With regards to the KCC2 co-transporter this story becomes more interesting as it may have an impact on febrile seizures and seizures in children as the levels of KCC2 are lower and so the Cl concentration dynamics are not regulated in the same manner as in the adult brain. This may lead to age-related differences in seizure susceptibility between children and adults. Given the results presented in this manuscript some brief discussion on this topic may help highlight the impact of this finding.

5. Given the amount of detail in the computational model there remains a number of network parameters that would be interesting to explore. Of specific interest would be the volume dynamics as there is ample experimental data demonstrating substantial changes in interstitial volume prior to seizure onset.

1. In its current form, the schematic in figure 1 gives the impression that vasculature and astrocytic interactions are included in the model. I believe it is not. It might be useful to drop those cartoons from the schematic to prevent confusion regarding what is specifically being modeled.

2. Please show specific examples of each activity type (resting state, tonic spiking, and bursting), in figure 2 when they are first described.

3. The stimulation of IN was said to result in a firing rate of 270Hz. It does not seem to be very realistic. Is there experimental evidence to justify such an increase in firing rate of IN neuron prior to SLE? If the model would be changed to get a lower firing rate would this affect the results?

4. Does blocking or reducing GABA-A conductance in the model without additional IN stimulation result in SLE? It is not clear if that is the case. If it does, this would be an interesting result to show or reference as it is a more one-to-one comparison with the experimental model of SLE.

5. In figure 3, a better comparison might be to apply the same or similar depolarization to the model PY as was done in the experiment. This could better highlight the match between the experimental and modelling results.

6. When discussing the bifurcation diagrams, it might be beneficial to discuss the types of bifurcations that occur at the specific transitions between activity regimes.

7. Did the authors consider the effects of HCO_3_^-^ in GABAergic currents? Since it was included as part of the GABA current and the fact the HCO_3_^-^ concentrations are crucial for maintaining GABA-A receptor reversal potentials some discussion on the role of HCO_3_^-^ in this regard would be needed.

8. Volume dynamics were included in this model and have been previously explored in other models (e.g., Schiff lab). How did volume change during the course of the simulation? How does volume impact the bifurcation diagrams for dendritic vs somatic K concentration?

*Reviewer #3 (Recommendations for the authors):*

Gentiletti et al. uses a computational model to investigate the mechanism underlying focal seizures. The small-network model consists of one interneuron and four pyramidal cells with various active and passive currents, and detailed ion concentration dynamics. The main focus of the study is to validate the hypothesis and previous results about the interneuronal origin of focal seizures. Specifically, in the model seizures are induced by stimulating the interneuron, which then raises the extracellular potassium ([K]o) levels, leading to high-frequency spiking in pyramidal cells. Detailed analysis of the pre-ictal, ictal, and post-ictal periods during seizures is also offered.

While technically sound, the study does not offer any major new innovations to warrant publication in *eLife*. I believe that the analysis presented is similar, in many ways, to previous modeling studies, but with a slightly different view. As detailed below, in some places it feels that the model parameters are selected so that a desired result is produced. The idea that interneurons can cause seizures by itself is not entirely new. There is extensive experimental (some cited by authors) and some modeling data supporting this hypothesis. Thus, I feel that the paper is more suitable for a specialized journal in computational neuroscience.

Following are some specific concerns about the study.

Figure 2 and related text: The experimental data shown for comparison is confusing. It is not clear if these are new experiments done for this study or data from three different previous studies are combined for comparison with the model. Another confusion about the experimental data is that in the beginning, it is mentioned that seizures are induced by bicuculine in the isolated guinea pig brain (Figure 3A), but later it is mentioned that "The strong preictal firing of the PY cells was artificially triggered by the injection of a steady depolarizing current via the intracellular recording electrode to analyze the intracellular firing correlates during the SLE (Gnatkovsky et al., 2008)." This gives the impression that the experimental traces shown for comparison are from three different experiments where the seizures are induced by different mechanisms, and are selected because they match what the model does.

Figure 6. In some places, it seems that the model is made to behave like the experiment. For example, in Figure 6, no reason is given for why the pyramidal cells are stimulated with 6 pA current. What would be the circumstances in the tissue or intact brain where the pyramidal cells would receive such input?

Lines 413 – 419: The paper claims that the previous modeling studies induced seizures by either stimulating pyramidal cells or increasing [K]o, whereas in this study, seizure is induced in the model by stimulating the interneuron. On the surface, this is true. But in reality, the approach adopted by this study is an indirect method of raising extracellular potassium of pyramidal cells. Stimulating interneurons leads to higher [K]o and hence seizures. This is also acknowledged by the authors. However, what is not clear is that why in the network only interneurons would receive strong depolarizing stimulus but not pyramidal cells. It should also be noted that the interplay between interneurons and pyramidal cells during seizures under normal [K]o in a modeling study was first reported by Wei et al. (2014) (PMID: 24671540).

Line 437-439: "This suggests that in our model, a change in the concentrations of either [K^+^]o or [Cl^-^]i was not sufficient to initiate an SLE and that an increase in both is necessary." This seems to be a model-specific effect because several modeling studies have shown that raising [K]o alone can cause the network to enter seizure-like state.

Line 250: [Na+]I should be [Na+]i.

[Editors’ note: further revisions were suggested prior to acceptance, as described below.]

Thank you for resubmitting your article "Focal seizures are organized by feedback between neural activity and ion concentration changes" for consideration by *eLife*. We apologise for the delay in assessing your revised manuscript, this was caused by our inability to secure one of the original reviewers which required finding an additional reviewer. This version of your article has now been reviewed by 3 peer reviewers, and the evaluation has been overseen by a Reviewing Editor and Ronald Calabrese as the Senior Editor. The following individual involved in the review of your submission has agreed to reveal their identity: Joseph V Raimondo (Reviewer #1).

In general, the reviewers were positive about the manuscript and the changes made.

Essential revisions:

We ask that you focus specifically on addressing the following

1. Clarify the specific novel contributions of this work in the abstract. The role of Na+, Na+/K^+^ pump on termination and Cl^-^ on seizure initiation have been previously shown.

2. The bifurcation analysis that suggests soma and dendrite differences for K^+^ concentration resulting in different regimes is interesting. Could you add to the discussion whether there is any experimental evidence for such significant differences in the concentrations for soma and dendrite?

3. With reference to the bicuculline model, could you add to the discussion in order to explain your mechanistic reasoning for how this “recruits the interneuronal network”

4. As the slowing of the interburst interval is one of the novel aspects of this work, it would be helpful to show that ion changes influence the inter-burst interval variations observed during the course of the seizure. Specifically, could you identify the bursting frequencies from the bifurcation analysis and confirm the role of Na^+^ accumulation in the slowing of burst interval.

Please find the full reviews below:

*Reviewer #1 (Recommendations for the authors):*

In this much revised manuscript the authors have improved several features of how ion dynamics were modelled enhancing the validity of their findings. The authors have gone to great lengths to address my concerns. In my opinion the manuscript enhances our understanding of the mechanisms underlying seizure dynamics.

Novelty

[Interneurons driving seizure onset] The authors have generated a thorough model of how intense activity of interneurons can drive K^+^ build-up and the initiation of seizures. This is certainly the most thorough and biophysically realistic model which recapitulates the electrographic feature of human seizures in a satisfying manner.

[Increased Na+/K^+^-pump driving the SLE termination and the post-ictal state]. The authors state that “We show for the first time that seizure termination and postictal state may be generated by the same mechanism mediated by increased activity of the Na+/K^+^-pump. It is an alternative mechanism of the postictal state, which previously has been suggested to depend on potassium undershoot (Krishnan and Bazhenov, 2011).” I disagree with the authors’ interpretation and insistence that this is novel. In Krishnan and Bazhenov, they explicitly describe essentially the same mechanism. Their potassium undershoot is caused by increased activity of the Na+/K^+^-pump (the show and state this explicitly).

“In our model extracellular potassium decays to baseline after an SLE offset, hence it cannot account for the postictal state.” This is also what (Krishnan and Bazhenov, 2011) show (see their Figure 8). I can’t see any fundamental difference between your interpretation and theirs, which is certainly satisfying, but not new as far as I can tell.

[Exponential IBI distribution]. I agree that the model satisfyingly shows how accounting for ion concentration changes can generate an exponential IBI distribution toward the end of the seizure.

Other responses:

I thank the authors for clarifying that in their model bicuculline does not block all of GABAergic inhibition, but is likely a transient perturbation which somewhat counterintuitively recruits the interneuronal network.

The authors have gone to great lengths to improve how Cl^-^ flux through GABAaRs and the Cl^-^ leak conductance was modelled. Figure 5 is a wonderful figure (along with Appendix – Figure 3). I am now satisfied that the ion dynamics were modelled in a suitable way which gives me much increased confidence in the authors’ findings.

*Reviewer #4 (Recommendations for the authors):*

In this work, the authors use a computational model to examine the role of ion dynamics in inhibition-mediated TLE seizures. Of significance, the seizure activity precisely matched the experimental data, specifically the time course of the inter-burst interval. The study also replicates previous experimental and computational observations on the role of K^+^ on seizure initiation, Na+, Na+/K^+^ pump on seizure termination. Further, the findings from this work suggest additional contributions of Na+/K^+^ pump to post-ictal depression and K^+^ influence on KCC2 pump promote seizure initiation.

While the paper’s findings are a significant contribution that emphasizes the importance of ion dynamics on seizure, the novel contribution highlighted by authors seems only as slight variations of previously proposed mechanisms. Nevertheless, the manuscript is definitely worthy of publication, perhaps in a more specialized journal.

Few suggestions that could improve the manuscript:

1. Bicuculline, a GABA antagonist induced the seizure in the experimental condition, but in the computational model, seizure was induced by increasing the inhibitory neurons’ activity. While the increase in inhibitory neuron’s activity is observed following bicuculline in the experiment, there is a missing network mechanism that results in this increase of inhibitory neurons activity following the application of bicuculline. It would be more compelling if the authors could identify this mechanism and demonstrate the onset of a seizure by bicuculline in the computational model.

2. It would be very helpful to clarify the specific novel contributions of this work in the abstract. The role of Na+, Na+/K^+^ pump on termination and Cl^-^ on seizure initiation have been previously shown. Also, it would be helpful ’or authors to note that K^+^ influenced the KCC2 pump time constant in Gonzalez et al., 2018.

3. The bifurcation analysis that suggests soma and dendrite differences for K^+^ concentration resulting in different regimes is interesting. It would be helpful to expand on this finding, and the report is one of the novel contributions. Is there any experimental evidence for such significant differences in the concentrations for soma and dendrite?

4. It would be helpful to show that ion changes influence the inter-burst interval variations observed during the course of the seizure. Specifically, the authors could identify the bursting frequencies from the bifurcation analysis and confirm the role of Na^+^ accumulation in the slowing of burst interval.

---

## [Author Response]

[Editors’ note: The authors appealed the original decision. What follows is the authors’ response to the first round of review.]

Comments to the Authors:We are sorry to say that, after consultation with the reviewers, we have decided that your work will not be considered further for publication by eLife.Although there was certainly enthusiasm about the model, which was found to be generally thorough and broadly satisfying in its behaviour and predictions, there was ultimately collective concern from all 3 reviewers about whether there was the requisite level of novelty and advance over prior work to justify publication in eLife.Please find the reviews below.Reviewer #1 (Recommendations for the authors):The model uses the NEURON framework and is well put together. On the whole its behaviour is both satisfying and reassuring. The model recapitulates the electrographic behaviour of animal model and human EEG recordings in particular the low-voltage fast activity preceding the tonic phase of a seizure, which transitions into bursting. The bursts then slow in frequency before postictal suppression of activity is observed. These dynamics emerge out of the model due to its inclusion of ion dynamics of K^+^, Na^+^ and Cl^-^ and multiple cellular mechanisms including accounting for excitatory and inhibitory cell populations, dendritic and somatic compartments, diffusion, glial buffering, ion channels and ion transporters.Whilst the work is very much worthy of publication, I am not convinced that it generates sufficiently novel findings and advances over previous work in the field (e.g. Krishnan 2011 and Krishnan 2015 and others) to be of sufficient interest to warrant publication in eLife. In my opinion the importance of K^+^, Cl^-^ and Na^+^ dynamics for seizure evolution have been demonstrated before. (e.g. increases in K^+^ driving SLE initiation, transition between tonic and bursting activity and seizure cessation occurring due to enhanced Na+/K^+^ ATPase activity).

We thank for overall positive judgment of our model. In our opinion our simulation data are novel and do advance mechanistic insight obtained from previous models as we demonstrate at least three important novel features/mechanisms that were not captured by any realistic computational seizure model so far.

Bazhenov’s team developed a model of neocortical seizures (Krishnan and Bazhenov 2011; Krishnan et al., 2015) which exhibit multiple transitions between tonic and bursting phases. Differently, our model considered for the first time a typical focal seizure pattern observed in human temporal lobe epilepsy (TLE) consisting of distinct phases, i.e., low voltage fast activity onset, tonic phase, clonic phase and postictal suppression phase. Human TLE seizures were shown to begin with increased firing of inhibitory interneurons (Elahian et al., 2018). Hence, the leading question behind our model was the mechanism by which interneuronal discharges initiate seizures. While this mechanism was investigated by others only once before (Gonzalez et al., 2018), we find these results not fully conclusive. In that model the direction of current flow through KCC2 was explicitly assumed to be outward and the influence of K^+^ concentration on flux direction was not considered. We note that in Gonzalez et al. (2018) rise in [K^+^]_o_ was larger than rise in [Cl^-^]_I_ (their Figure 4A) suggesting that in their model the direction of KCC2 transport would presumably be reversed. Therefore, the conclusion provided by the model, that SLE can be induced by K^+^ accumulation via outward K-Cl cotransport is still open for discussion. Our present model suggests an alternative scenario in which initial build-up of extracellular K^+^ occurs via voltage-gated K^+^ channels activated by intense interneuronal spiking. Our preliminary study showed seizure initiation by interneurons (Gentiletti et al., 2017) in a simple two-cell model, and the underlying mechanism was not investigated.

We show for the first time that seizure termination and postictal state may be generated by the same mechanism mediated by increased activity of the Na^+^/K^+^-pump. It is an alternative mechanism of postictal state, which previously has been suggested to depend on potassium undershoot (Krishnan and Bazhenov, 2011). In our model extracellular potassium decays to baseline after an SLE offset, hence it cannot account for the postictal state. It is important to mention that both patterns of postictal potassium time course were observed experimentally (with undershoot, e.g., Dreier and Heinemann (1991) and with slow decay e.g., Fisher et al., 1976). Our novel hypothesis, which is consistent with the experimental evidence that inhibition of Na^+^/K^+^-pump prolongs SLE and reduces post-SLE period (Haas and Jefferys, 1984), has never been demonstrated by any computational model so farThe model predicts for a first time that ion concentration changes at the end of seizures lead to exponential inter-burst interval (IBI) distribution as indeed observed experimentally. Recently, there is increased interest in the IBI pattern due to seizure taxonomy developed by Victor Jirsa’s group (Jirsa et al., 2014, Saggio et al., 2020). His generic epileptor model predicts logarithmic relationship of IBI which is in contrast to human seizure data analyzed by us in the manuscript and others (Bauer et al., 2017).

(1) I am concerned as to the applicability of the experimental data to this model. SLEs in the whole guinea pig brain were elicited using bicuculline, which blocks GABAaR transmission yet current from the IN to pYs via activated GABAaRs is presumably an important component of the computational model?

We thank the Reviewer for raising this point as it allows us to better clarify this issue. We tend to maintain our opinion that choice of experimental model was appropriate. On the other hand, we recognize that our original presentation needed further explanation. We used experimental data from SLEs induced by brief perfusions of convulsive drugs in the whole guinea pig brain in vitro model (Gnatkovsky et al. 2008; Uva et al. 2009; Uva et al. 2015). In the whole brain in vitro model, transient (3 minute) arterial perfusion of 50 microM bicuculline reduces GABAergic inhibition only transiently to 60-70% (Gnatkovsky et al., 2008, Figure 1). This treatment does not block GABAergic transmission, but reduces it. As discussed in Gnatkovsky et al. (2008), this brief bicuculline application likely affects interneuron-interneuron inhibition more than interneuron-principal cell inhibition and recruits the interneuronal network. With this treatment (and also with 5-minute perfusion with 100 µM 4 aminopyridine) SLE onset correlates with interneuron bursting and with no activity in principal cells (Uva et al., 2015). The same phenomena have been observed in spontaneous seizures in patients with mesial-temporal epilepsy (Elahian et al., 2018). Hence we choose bicuculline model to guide our modeling work because: (i) SLEs in this model are initiated by enhanced firing of inhibitory interneurons, (ii) SLE pattern in this model closely resemble human temporal lobe seizure (iii) LFP, single unit and ionic data are available. We explain the choice of this model in the Introduction (ln. 64-69) and we describe it more thoroughly in the Results (ln. 98-103).

(2) Understandably the model cannot recapitulate all biological detail but there are some aspects about the model which seem odd and would need justification:a. e.g. Intracellular HCO_3_^-^ is higher than intracellular bicarbonate (15 mM) being higher than extracellular bicarbonate (11 mM). This suggests that intracellular pH is more alkaline than extracellular pH which is not the case. In addition, this would make the bicarb reversal +8 mV which is high and unrealistic.

In the original model the [HCO3]_o_ was calculated based on electric and osmotic equilibrium but we admit that its value was not optimal. In the revised version of the model we modified some ionic concentrations and we set [HCO3]_o_ = 25 mM, [HCO3]_i_ = 15 mM (Doyon et al., 2011) giving the HCO_3_^-^ reversal potential, E_HCO3_ = -13.4 mV. It is described in the *Initial ion concentrations* subsection of the revised manuscript starting at ln. 878.

b. In the neurons the Na^+^ leak conductance is 40% of the K^+^ leak conductance, this seems very high and not a typical ratio of permeabilities for neurons.

In the revised model, Na^+^ leak current conductance was determined by a condition that at rest all Na^+^ and K^+^ currents are in ratio -3/2 and are counterbalanced by a Na^+^/K^+^-pump. Accordingly, there is no net flux of Na^+^ and K^+^ ions at rest. E.g, in the PY soma it led to the following values: *g_K,leak_ =* 3*10^-5^ S/cm^2^, *g_Na,leak_ =* 1.5*10^-5^ S/cm^2^. These values are comparable to g_Na,leak_/g_K,leak_ ratio used in other realistic models e.g., 45% (Krishnan et al., 2015), 49 % (Wei et al., 2014b). We note that leak currents in our manuscript are calculated using ohmic model with conductance values expressed in unit of S/cm^2^, while permeability values are typically expressed in units of m/s and are used in Goldman-Hodgkin-Katz model. Therefore, leak conductance values and permeability values are difficult to compare directly. In order to validate our setting of leak conductances we calculated resting membrane potential in two ways. First, we used classical Goldman-Hodgkin-Katz voltage equation and second, we calculated resting membrane voltage derived from the ohmic model i.e., Vrest = (g_K_E_K_ + g_K_E_K_ + g_Cl_E_Cl_)/(g_K_ + g_Na_ + g_Cl_). For PY somatic compartment, the leak conductances used in our model were: *g_K,leak_ =* 3*10^-5^ S/cm^2^, *g_Na,leak_ =* 1.5*10^-5^ S/cm^2^, *g_Cl,leak_ =* 1*10^-5^ S/cm^2^. For these values we obtained an agreement with the Goldman-Hodgkin-Katz model having permeabilities P_K_ : P_Na_ : P_Cl_ = 1 : 0.075 : 0.1. These permeabilities are comparable to experimental data (e.g., in the squid giant axon the ratio is 1: 0.03: 0.1) what gave us confidence that the leak current conductances used in the model are in physiological range.

c. It wasn't clear whether the leak conductances for the various ions were actually contributing to the ion dynamics, e.g. was Cl^-^ flux through the baseline Cl^-^ leak conductance in neurons contributing to changes in [Cl^-^]i?

In the revised model we increased *g_Cl,leak_* and we set E_Cl_ > E_k_ at rest, hence the KCC2 was not in equilibrium. In this way all leak conductances were contributing to the ion dynamics, including baseline Cl^-^ flux, which at rest was compensated by Cl^-^ extrusion by KCC2.

d. Ek = Ecl at rest, this is also not physiological, I understand why this was done so that KCC2 was at equilibrium at baseline, but this is not ideal. Rather there should be a tonic Cl^-^ leak influx ensuring that "at baseline" Ecl>Ek as observed experimentally.

We thank the Reviewer for this suggestion. We modified the model accordingly.

(3) Modelling of Cl^-^ changes especially Cl^-^ flux through GABAaRs. I am also not entirely sure that Cl^-^ flux through GABAaRs was modelled correctly to capture the potential for biphasic/ depolarizing responses via intensely activated GABAaRs (the authors should note amongst others Ruusuvuori 2004). The calculation of Egaba included HCO_3_^-^ (line 833) but how was Icl (Cl^-^ flux) via activated GABAaRs calculated? Ie in the model if Vm is at the GABAaR reversal potential (Egaba) would Cl^-^ ion flux into the cell via GABAaRs (Icl) be zero? Ie in the model if Icl calculated as 4/5 of Igaba and Igaba = Ggaba(Vm – Egaba), then when Vm = Egaba, Igaba and consequently Icl is 0. This shouldn't be the case. Rather Igaba = Icl + Ihco3 where Icl = gcl(Vm – Ecl) and Ihco3 = ghco3(Vm – Ehco3) and ggaba = 4/5xgcl +1/5xghco3. Seeing as Cl^-^ accumulation is a fundamental part of the model this should have been made more clear. E.g. in Figure 5, could Icl via GABAaRs also be plotted? I worry as the Cl^-^ influx in the model seemed to be coming predominantly via KCC2 (due to the raised extracellular K^+^) whereas experimentally this is likely also coming predominantly through GABAaRs. This is reflected in the time course of [Cl^-^]i changes in the model which are slower than [K^+^]o and continue increasing until the end of the SLE which is not typically what is observed experimentally (see intracellular Cl^-^ recordings e.g. Raimondo 2013).

These are very relevant points and we recognize that we used simplified formula for Icl flux through GABAaR in the original manuscript. In the revised manuscript we carefully addressed this issue and corrected our approach. We divide our answer into separate sections.

Modelling of Cl^-^ flux through GABAaRs

Indeed, in the original manuscript we considered HCO_3_^-^ influence on the I_Cl_ GABAA current via E_GABAa_, which depended on both Cl^-^ and HCO_3_^-^ concentrations but we didn’t model I_HCO3_ flux separately. Following Reviewer’s suggestion, we used extended formula of Igaba as described by the Reviewer and also used by Jedlicka et al., (2011):

Igaba = Icl + Ihco3

Icl = (1-P)*g*(v-Ecl)

Ihco3 = P*g*(v-Ehco3)

Egaba = P*Ehco3 + (1-P)*Ecl

P = 0.18 (relative permeability)

HCO3i = 15 mM

HCO3o = 25 mM

With the modified formula for Igaba we obtained model behavior that was qualitatively the same as in the original model. It shows that our conclusions were not affected by a simplified formula for Igaba or by imprecise setting of [HCO_3_^-^]_o_. However, we agree that the recommended changes were appropriate, and we thank the Reviewer for this suggestion.

Cl^-^ influx through GABAaR and KCC2

In the revised manuscript we plot Cl^-^ currents flowing via GABAaR and KCC2 (Figure 5B, two bottom panels). It can be seen that during an SLE chloride accumulation is dominated by Cl^-^ influx via GABAaR with smaller flux via KCC2 transporter and only small contribution from I_Cl,leak_.

Continuous increase of [Cl^-^]_i_ during an SLE

Indeed, in our model, the [Cl^-^]_i_ continues to increase due to both GABAaR and KCC2 mediated currents until the end of an SLE. In other realistic models (Krishnan and Bazhenov, 2011; Krishnan et al., 2015), which mimic neocortical electrographic seizures, [Cl^-^]_i_ also builds up over the course of the whole SLE and the [Cl^-^]_i_ increase was proposed to contribute to seizure termination by changing the minimal [K^+^]_o_ level required to initiate next tonic spiking episode (Krishnan and Bazhenov, 2011). These modeling results showing continuous [Cl^-^]_i_ increase during SLE agree with [Cl^-^]_i_ recordings during SLE induced in vivo by 4-AP (Sato et al., 2017; Figure 8). On the other hand, they differ from experimental data showing decline of [Cl^-^]_i_ during SLE induced by bath application of Mg^2+^-free solutions in rat organotypic hippocampal slice cultures or acute mice cortical slices (Raimondo et al., 2013; Figure 3 and 4). These findings might suggest that there is no single scenario of [Cl^-^]_i_ time course during SLE and that different seizure models could exhibit different [Cl^-^]_i_ evolution patterns, which depend on simultaneously changing [K^+^]_o_ among other mechanisms. E.g., it has been shown that there is differential involvement of inhibitory interneurons in 4AP and low Mg^2+^ models (Codadu et al., 2019), which, in turn, could lead to different K^+^ and Cl^-^ accumulation patterns. We discuss this issue in the revised manuscript (ln. 518 – 525).

Biphasic/ depolarizing responses following GABAaR activation

The model has a potential to generate biphasic/ depolarizing responses via intensely activated GABAaRs (Kaila et al., 1997; Ruusuvuori et al., 2004; Viitanen et al., 2010). These responses could be elicited in the model only if GABAergic inputs were located in small compartments representing distal dendrites (i.e., having smaller volume and higher GABAA receptor density than soma in the reference model). An example of hyperpolarizing-depolarizing GABAA response in a small dendritic compartment is shown in Appendix I – figure 3. The depolarizing component is related to strong Cl^-^ accumulation and depolarizing shift in *E_GABAa_*. We hypothesize that depolarizing responses following GABAaR activation might be more readily observed due to activation of dendrite-targeting somatostatin-expressing (SST) interneruons rather than soma-targeting parvalbumin-positive (PV) interneurons. In our reference model GABAA synapses are located in the soma, which is based on the observation that activation of PV interneurons was implicated in spontaneous seizures (Toyoda et al., 2015). Due to large size of somatic compartment, GABAaR activation leads to moderate Cl^-^ accumulation giving rise to limited shift in *E_GABAa_* (Figure 5B)*_._* Our hypothesis is in agreement with other studies suggesting that [Cl^-^]_i_ can change rapidly and contribute to depolarizing GABAA responses especially in the structures with low volume to GABAA receptor density ratio (Staley et al., 1995; Staley and Proctor, 1999).

(4) Perhaps the authors could make it more clear that the exact same experimental data was also presented in Gentilleti 2017.

We thank the Reviewer for pointing this out. We added appropriate references to the experimental data.

Reviewer #2 (Recommendations for the authors):Gentiletti et al. investigated potential role of non-synaptic mechanisms driving seizure-like event (SLE) generation. Using a detailed computational model of a small network of neurons, the authors demonstrate that the complex interaction between specific ion species may give rise to SLE. The detailed analysis of the computational model provides an interesting approach to developing a unifying framework for neural dynamics. The strength of this manuscript is in the direct validation of key aspects of the computational model using in vitro electrophysiology. Additionally, predictions made by the model such as the slowing of inter-burst intervals are subsequently validated in both human and mouse data. The conclusions made by the authors of this manuscript are supported by their results and are in line with previous work in the field. Finally, the manuscript does a good job relating the novel results of this new manuscript with established results in the field.

We thank for overall positive judgment of our model and our results.

1. A strength of this manuscript is the direct validation of the computational model with experimental results. Given the multitude of methods available for inducing seizure-like events (SLE) in vitro, it is a bit surprising that the authors chose to use an arterial application of bicuculline (Figure 3). Bicuculline is a competitive antagonist of the GABA-A receptor resulting in the reduction of inhibitory GABAergic signaling. However, the SLE induced in the model is caused by an increase in inhibitory activity, through direct depolarization, rather than a decrease in GABAergic signaling. At first glance these methods for inducing SLE seem to be at odds with one another. Given the observed increase in extracellular K^+^ at SLE onset, this mismatch in the method for SLE generation in vitro and in silico may further highlight one of the primary claims of the manuscript specifically that disruption of ionic homeostasis rather than solely synaptic excitatory/inhibitory imbalance is a mechanism for SLE generation. Additional discussion of this observation and similar phenomena in other methods for SLE generation in vitro would further strengthen this interesting point.

We thank the Reviewer for this relevant comment. We agree that the rationale for using direct depolarization of the interneuron to initiate an SLE and the use bicuculline model data could seem inconsistent and needed further explanation. Increased interneuron discharges and decreased PY activity around the time of seizure onset have been shown in SLE induced in vitro by 4-AP (Ziburkus et al., 2006; Lévesque et al., 2016; Uva et al., 2015) and brief bicuculline applications (Gnatkovsky et al., 2008), in acute and chronic seizure models in vivo (Grasse et al., 2013; Miri et al., 2018; Toyoda et al., 2015) and temporal lobe of epilepsy patients (Elahian et al., 2018). Motivated by this general observation obtained under different experimental conditions, we aimed to test whether fast intraneuronal discharges can indeed trigger an SLE *in silico* and if so, what would be the underlying mechanism.

We used data obtained during transient bicuculline perfusion in the isolated guinea pig brain because SLE in this model closely resemble human TLE seizures (de Curtis and Gnatkovsky 2009) and are initiated by increased interneuron discharges (Gnatkovsky et al., 2008). Brief bicuculline application likely affects interneuron-interneuron inhibition more than interneuron-principal cell inhibition (Gnatkovsky et al., 2008). With this treatment SLE onset correlates with interneuron bursting and with no activity in principal cells (Uva et al., 2015), suggesting that interneuron-principal cell GABAergic signaling is still functional at the beginning of an SLE. On the other hand, reduced inhibition among IN cells could possibly produce interictal spikes generated by interneurons leading to increase in extracellular potassium, that in turn would further depolarize interneuronal network and initiate SLE (de Curtis and Avoli, 2016, Figure 4). In alternative scenario reduced inhibition among interneurons could lead to abrupt switch from low firing to high firing rate mode of IN network (Rich et al., 2020). Hence, to initiate an SLE we decided to simulate only the ‘effect’ of reduced interneuron-interneuron inhibition manifested by their increased discharge. In this way we made our model more general and applicable to numerous experimental data in which paradoxical increase of GABAergic cell firing is observed at seizure onset. We explain choice of bicuculline model in the Introduction (ln. 64-69) and we describe bicuculline action more thoroughly in the Results (ln. 98-103).

2. Bifurcation analysis in figure 4A produces interesting results that are in line with and supported by previous work. As stated in the text, an assumption made by the authors is that the intracellular dendritic and somatic Na^+^ is equal. The authors further mention that this assumption may result in an overestimation of dendritic Na^+^ and Na/K pump activity but do not discuss how this might impact the results presented in the bifurcation diagrams. Please include.

Following Reviewer’s suggestion, in Figure 4A we present more detailed bifurcation analysis with separate values for dendritic and somatic [Na^+^]_i_ in the revised version of the manuscript.

3. It is striking that the time course of the extracellular dendritic K^+^ concentration is much slower than for the soma. It is not clear if the delayed increase in dendritic K^+^ is a prediction of the computational model or if it has been experimentally observed and incorporated into the model as such. Some discussion clarifying this point is needed. To that point, are the authors suggesting that the longitudinal diffusion of K^+^ from soma to dendrite is driving the delayed increase? If so, how might the observed dynamics change if the direction was reversed? The computational model contains an inhibitory neuron which seems to target specifically the soma. For this reason, it is not surprising that the somatic K^+^ increases first. However, peri-somatic inhibition is a characteristic of PV-inhibitory interneurons. Somatostatin (SOM) expressing inhibitory interneurons predominantly target dendrites. Given recent studies showing that stimulation of either SOM or PV interneurons can trigger seizure onset, how might this impact the bifurcation dynamics presented here?

We thank the Reviewer for pointing out this important issue and we agree that it was not properly discussed. Indeed, the time course of [K^+^]_o_ rise is slower and its magnitude is diminished in dendritic vs. somatic compartment. We are not aware of any experimental data showing such effect; hence it can be considered a model prediction.

The Reviewer is correct that initial fast rise of [K^+^]_o_ in somatic compartment and slower rise in dendritic segment is related to inhibitory neuron, which targets specifically the soma and also to small longitudinal diffusion between extracellular compartments. During the course of the SLE, as interneuron activity decreases, the additional contribution to [K^+^]_o_ comes from pyramidal cell firing, The dendritic conductance of voltage-gated K^+^ currents associated with spiking is about 10% of the somatic conductance (Fransen et al., 2002) hence release of K^+^ ions into the dendritic interstitial space is smaller than in somatic compartment. On the other hand, radial diffusion (to the bath) and glial buffering processes have the same efficiency in both compartments maintaining higher K^+^ near the soma.

As the Reviewer points out, indeed, activation of either PV or SOM inhibitory interneurons can trigger SLE (Yekhlef et al., 2015). In our model, we predict that if the interneuron was targeting the dendrite, it could still trigger an SLE. As shown by the bifurcation diagram in Figure 4A (first panel) an increase in either somatic or dendritic [K^+^]_o_ leads to transition from rest to tonic spiking and then bursting. Given these two possible scenarios, our choice to include soma-targeting fast spiking interneuron was motivated by the fact that PV interneurons were more often selected as target for optogenetic stimulation (Ellender et al., 2014; Sessolo et al., 2015) and were supposed to be implicated in spontaneous seizures (Toyoda et al., 2015). We addressed this relevant issue in ln. 527-535 and ln. 543-548 of the revised manuscript.

4. The results pertaining to Cl concentration are interesting and extend a large number of recent studies examining the role of Cl in seizure dynamics. With regards to the KCC2 co-transporter this story becomes more interesting as it may have an impact on febrile seizures and seizures in children as the levels of KCC2 are lower and so the Cl concentration dynamics are not regulated in the same manner as in the adult brain. This may lead to age-related differences in seizure susceptibility between children and adults. Given the results presented in this manuscript some brief discussion on this topic may help highlight the impact of this finding.

Indeed, KCC2-dependent chloride and potassium homeostasis may depend on age. E.g., it has been shown in rats that KCC2 shows little expression in neonates and increases with neuronal maturation but there is also high heterogeneity of KCC2 expression in pyramidal cells at birth, with some neurons having efficient KCC2 activity and low [Cl^-^]_i_ (Ben Ari, 2002). With regard to febrile seizures, it has been shown that they may be related to reduced Cl^-^ extrusion through KCC2 (Puskarjov et al., 2014) what in turn could lead to decreased formation of dendritic spines. During an SLE generated in our model, *E_K_* is higher than *E_Cl_* and KCC2 operates in reverse direction (Figure 5). Despite KCC2 activity leads to Cl^-^ and K^+^ transport into the cells during an SLE and contributes to positive *E_Cl_* shift, depolarizing GABAA responses are not observed in the model. In the revised manuscript we discuss that limited shift in EGABAa in the model may be related to large size of the somatic compartment, which is targeted by inhibitory input from the IN (ln. 465-680 and Appendix I – figure 3). At the same time, we would prefer not to relate our model findings to neonatal seizures, as the current model may be too limited (e.g., lack of NKCC1 cotransporter) to mechanistically account for these phenomena.

5. Given the amount of detail in the computational model there remains a number of network parameters that would be interesting to explore. Of specific interest would be the volume dynamics as there is ample experimental data demonstrating substantial changes in interstitial volume prior to seizure onset.

This is relevant point and indeed we recognize that we didn’t discuss this topic sufficiently in our original presentation. In the revised manuscript we added Appendix I – figure 4 showing volume changes. It can be seen that extracellular space (ES) volume shrinks maximally by about 27%, which is comparable to Dietzel et al. (1980) who reported average reduction of the ES by about 30%. Intracellular space (IS) volume expands about 4%, which comes from the fact that the total volume is constant (ES+IS = const). Volume changes affect all ion concentrations as demonstrated using the time course of the intracellular A^-^ and extracellular Cl^-^ ion concentrations in Appendix I – figure 4. We selected these ions as an example because [A^-^]_i_ concentration doesn’t depend on any other mechanism while [Cl^-^]_o_ is not regulated by the efficient Na^+^/K^+^ pump and glial buffering. Despite extracellular volume changes in our model are significant, counterintuitively, these changes don’t have big impact on the observed model dynamics. We notice that due to slow time course of the volume changes, extracellular concentrations of all ions are efficiently compensated by other mechanisms e.g., radial diffusion to the bath, which act on a faster time scale. Additionally, extracellular potassium is under efficient and relatively fast control of the glial buffering and Na^+^/K^+^ pump which dominate the effect of extracellular volume changes in both somatic and dendritic compartments. However, it should be noted that in our model glial cell swelling was not included and diffusion to the bath was not affected by the volume changes. In the revised manuscript we mention it as one of the model limitations. In the real tissue astrocyte swelling may be significant and flow of ions and oxygen (affecting Na^+^/K^+^-pump activity) may be reduced contributing to seizures and spreading depression (Hubel and Ullah, 2016) (ln. 865-875).

Regarding preictal volume changes indeed, constriction of blood vessels (Patel et al., 2013), lowering extracellular ca^2+^ concentration was observed. The exact mechanism linking these changes to spontaneous seizure generation is still unclear but might be investigated with computational models. As discussed in Somjen (2004; p. 84), lowered concentration of [ca^2+^]_o_ might decrease surface charge screening effect and depolarize cells leading to network hyperexcitability. However, the effect of surface charge on membrane potential was not included in the current model hence it is not well-suited to explore this question.

6. In its current form, the schematic in figure 1 gives the impression that vasculature and astrocytic interactions are included in the model. I believe it is not. It might be useful to drop those cartoons from the schematic to prevent confusion regarding what is specifically being modeled.

Indeed, vasculature is included in the model only indirectly by fluxes to/from the bath medium, which represents various processes such as diffusion to more distant areas of the brain and cerebrospinal fluid, active transport of potassium into capillaries and potassium spatial buffering by astrocytes. Glial cell interactions are not included explicitly but the process of spatial potassium buffering by astrocytes is accounted for by the glial buffer included in each extracellular compartment. We agree that as the vasculature was not explicitly modeled, the blood vessel drawing might create confusion and should be removed. On the other hand, we would prefer to keep astrocyte cells in the model diagram to emphasize that potassium regulation by glial cells is essential part of the model.

7. Please show specific examples of each activity type (resting state, tonic spiking, and bursting), in figure 2 when they are first described.

We agree that the details of membrane potential traces are not well visible in Figure 2, but showing them as insets in that figure is also not optimal due small panel size. In the revised manuscript we clarify (ln. 113-114) that the three types of activity are shown in an extended time scale in Appendix I – figure 3.

8. The stimulation of IN was said to result in a firing rate of 270Hz. It does not seem to be very realistic. Is there experimental evidence to justify such an increase in firing rate of IN neuron prior to SLE? If the model would be changed to get a lower firing rate would this affect the results?

As mentioned in Gnatkovsky et al. (2008), putative interneurons in the entorhinal cortex generated fast discharges above 150 Hz before or at the onset of an SLE. Although some interneurons were indeed able to generate very fast (e.g., 270 Hz) discharges, the mean firing rate was around 166 Hz (Uva et al., 2015). Therefore, in the revised manuscript the initial firing rate of the interneuron was 150 Hz in order to match these experimental data.

9. Does blocking or reducing GABA-A conductance in the model without additional IN stimulation result in SLE? It is not clear if that is the case. If it does, this would be an interesting result to show or reference as it is a more one-to-one comparison with the experimental model of SLE.

Blocking GABA-A conductance in the model doesn’t lead to SLE by itself. It might sound counterintuitive as in general blockade of inhibition was shown to lead to synchronized paroxysmal activity (Connors, 1984). Several studies demonstrated that prolonged perfusion of brain slices with bicuculine generates large interictal spikes and afterdischarges lasting max 1 second (see de Curtis and Avanzini 2001), but never leads to true focal SLEs. As discussed in the answer to Reviewer #2 point 1 above, the mechanism leading to SLE after transient reduction of GABAergic inhibition is not entirely clear, but it may not be primarily related to a reduction of IN to PY inhibition but rather to IN disinhibition. We might hypothesize that in order to fully investigate this effect using a model one should include more extended interneuronal network with mutual inhibitory interactions. Blocking or decreasing GABA-A conductance in such a network could disinhibit IN cells and produce interictal spikes generated by interneurons contributing to increase of extracellular potassium. It would further depolarize interneuronal network and initiate SLE (de Curtis and Avoli, Figure 4). In an alternative scenario, reduction of GABA-A conductance in IN cells could lead to abrupt switch from low firing to high firing rate mode of the intraneuronal network (Rich et al., 2020) and trigger SLE via [K^+^]_o_ and [Cl^-^]_i_ accumulation.

10. In figure 3, a better comparison might be to apply the same or similar depolarization to the model PY as was done in the experiment. This could better highlight the match between the experimental and modelling results.

Thank you for your remark. We agree that indeed PY cell membrane potential trace was misleading. We use alternative pyramidal cell data in Figure 3 of the revised manuscript.

11. When discussing the bifurcation diagrams, it might be beneficial to discuss the types of bifurcations that occur at the specific transitions between activity regimes.

Indeed, to get a better theoretical insight into transitions between different firing regimes it would be useful to analyze types of bifurcations associated with these transitions. Firstly, we note that in a model as complex as ours (each 2-compartment cell is described by about 60 differential equations) it would require a separate project to be carried out using Content or XPP Auto software package. Secondly, we used Neuron simulator, in which some differential equations are built-in in the simulator and while their general form is known, obtaining all necessary details is not straightforward. Finally, we prepared our manuscript for a wide audience of physiologists or clinicians working in the field of epilepsy. To increase clarity, we decided to avoid concepts from other disciplines e.g., dynamical system theory. Nevertheless, we appreciate your suggestion as it may motivate us to complement our current work with the future study directed to more theoretically oriented audience.

12. Did the authors consider the effects of HCO_3_^-^ in GABAergic currents? Since it was included as part of the GABA current and the fact the HCO_3_^-^ concentrations are crucial for maintaining GABA-A receptor reversal potentials some discussion on the role of HCO_3_^-^ in this regard would be needed.

In the original model we included HCO_3_^-^ only in the calculation of the GABAA reversal potential. Following Reviewers suggestion in the revised manuscript we used an extended formula of Igaba with explicitly modeled HCO_3_^-^ currents (Jedlicka et al., 2011):

Igaba = Icl + Ihco3

Icl = (1-P)*g*(v-Ecl)

Ihco3 = P*g*(v-Ehco3)

Egaba = P*Ehco3 + (1-P)*Ecl

P = 0.18 (relative permeability)

HCO3i = 15 mM

HCO3o = 25 mM

In our reference model, GABAA receptors are located in the relatively large somatic compartments, what limits [Cl^-^]_i_ accumulation and prevents depolarizing GABAA responses even when HCO_3_^-^ currents are included explicitly. However, the role of HCO_3_^-^ ions becomes especially evident in the simulation of GABAA responses in small dendritic compartments, in which [Cl^-^]_i_ accumulation together with depolarized E_HCO3_ leads to depolarizing shift in *E_GABAa_* as shown in Appendix I – figure 3.

13. Volume dynamics were included in this model and have been previously explored in other models (e.g., Schiff lab). How did volume change during the course of the simulation? How does volume impact the bifurcation diagrams for dendritic vs somatic K concentration?

As discussed in the answer to Reviewer #2 point 5 above, volume changes were of realistic magnitude but didn’t have big impact on the observed model dynamics. Changes related to shrinkage of extracellular space had slower dynamics than other homeostatic processes and were efficiently compensated by various mechanisms (diffusion, glial buffering and transmembrane ion transporters). On the other hand, as shown by Hubel and Ullah (2016) astrocyte swelling may play a significant role leading to reduced flow of ions and oxygen (affecting Na^+^/K^+^-pump activity) in the tissue and contributing to seizures and spreading depression. These effects (reduced diffusion and reduced Na^+^/K^+^-pump rate) were not included in our model what is described as one of the model limitations (ln. 865-875).

Reviewer #3 (Recommendations for the authors):Gentiletti et al. uses a computational model to investigate the mechanism underlying focal seizures. The small-network model consists of one interneuron and four pyramidal cells with various active and passive currents, and detailed ion concentration dynamics. The main focus of the study is to validate the hypothesis and previous results about the interneuronal origin of focal seizures. Specifically, in the model seizures are induced by stimulating the interneuron, which then raises the extracellular potassium ([K]o) levels, leading to high-frequency spiking in pyramidal cells. Detailed analysis of the pre-ictal, ictal, and post-ictal periods during seizures is also offered.While technically sound, the study does not offer any major new innovations to warrant publication in eLife. I believe that the analysis presented is similar, in many ways, to previous modeling studies, but with a slightly different view. As detailed below, in some places it feels that the model parameters are selected so that a desired result is produced. The idea that interneurons can cause seizures by itself is not entirely new. There is extensive experimental (some cited by authors) and some modeling data supporting this hypothesis. Thus, I feel that the paper is more suitable for a specialized journal in computational neuroscience.

We thank the Reviewer for careful reading of our manuscript and his/her pertinent remarks. We fully agree that the idea that interneurons can cause seizures is not new but to the best of our knowledge the underlying mechanism is still a matter of debate. Specifically, the relative contribution of raised extracellular potassium (Avoli et al., 1996; Gnatkovsky et al., 2008) vs. increased intracellular chloride leading to depolarizing GABA responses (Lillis et al., 2012; Alfonsa et al., 2012) remains to be established. Furthermore, the question regarding the main source of extracellular potassium rise (voltage-gated K^+^ currents associated with interneuronal spiking vs. K^+^ extrusion via KCC2) is still open. We address and provide answers to these questions in our *in silico* study. We are convinced that such theoretical explanations contribute to better understanding of these complex mechanisms, which are currently not fully understood.

As also argued in response to Reviewer #1 our model exhibits at least three novel features/mechanisms that were not captured by any realistic computational seizure model so far. (i) The model reproduces all four phases of focal seizures (low voltage fast activity onset, tonic phase, clonic phase and postictal suppression phase) as observed in human focal epilepsy. (ii) It suggests a novel, Na^+^/K^+^-pump-based link between the seizure termination mechanism and postictal suppression state. (iii) It shows that ion accumulation-related mechanism may lead to specific scaling law of inter-bursting intervals (IBI) observed at the end of seizures. The exponential scaling law predicted by the model has been verified experimentally. We note that the recognized epileptor model developed by Victor Jirsa’s group (Jirsa et al., 2014, Saggio et al., 2020) predicted logarithmic relationship of IBI which is in contrast to our model and human seizure data. Therefore, we hope that our work provides satisfactory amount of novel contribution and offers new and advanced mechanistic insight into unsettled or unexplored seizure generation processes to warrant publication in *eLife*.

Following are some specific concerns about the study.Figure 2 and related text: The experimental data shown for comparison is confusing. It is not clear if these are new experiments done for this study or data from three different previous studies are combined for comparison with the model. Another confusion about the experimental data is that in the beginning, it is mentioned that seizures are induced by bicuculine in the isolated guinea pig brain (Figure 3A), but later it is mentioned that "The strong preictal firing of the PY cells was artificially triggered by the injection of a steady depolarizing current via the intracellular recording electrode to analyze the intracellular firing correlates during the SLE (Gnatkovsky et al., 2008)." This gives the impression that the experimental traces shown for comparison are from three different experiments where the seizures are induced by different mechanisms, and are selected because they match what the model does.

We thank the Reviewer for pointing not sufficient information on experimental data used in Figure 3a and we recognize that our original description needed more clarity. Some experimental recordings used for comparison with model simulation were published previously, specifically, LFP and interneuron data were published in Gnatkovsky et al., 2008 (Figure 3A and 6B) and Gentiletti et al., 2017 (Figure 3). The PY cell and [K^+^]_o_ data have never been published before. It is challenging to carry out simultaneous intracellular recording from pyramidal cell and interneuron together with extracellular potassium recording in the in vitro isolated whole brain. Hence, the four signals presented in Figure 3a were recorded in different experiments during SLE in the entorhinal cortex of the isolated guinea pig brain, induced by a brief and transient 3-min perfusion of 50 microM bicuculline (that reduces GABAergic transmission by about 40%).

We would also like to clarify that the experimental traces shown for comparison were not selected to match the model simulations. Rather these experimental data were used as guidelines to create and evaluate the computational model.

Finally, we thank the Reviewer for the comment regarding the confusion created by preictal firing of the PY cell. We agree that indeed the PY cell membrane potential trace was misleading. We use alternative pyramidal cell data in Figure 3 of the revised manuscript.

Figure 6. In some places, it seems that the model is made to behave like the experiment. For example, in Figure 6, no reason is given for why the pyramidal cells are stimulated with 6 pA current. What would be the circumstances in the tissue or intact brain where the pyramidal cells would receive such input?

The main reason for pyramidal cell stimulation is given in the manuscript text and legend to Figure 6 (“The absence of excitatory dendritic synaptic input was compensated with a steady depolarizing DC current […] adjusted to preserve the original duration of the SLE”) but we are glad to use this opportunity to further explain why such manipulation was necessary. In order to observe inter-burst intervals (IBI) evolution and find best fit of a mathematical function we removed the stochastic component, i.e., background input noise, from the model. Without this excitatory dendritic synaptic input, the SLE were shorter and there were less data for fitting. To restore SLE duration as was seen in reference simulation (Figure 2) we compensated the absence of excitatory noise input by a small depolarizing current. In the tissue or intact brain, the pyramidal cells could receive such steady input by means of optogenetic manipulation. On the other hand, it is not possible to remove stochastic component in the real tissue. Hence ‘noise-free’ experiments with compensative steady current injection can indeed be carried out only *in silico*.

Lines 413 – 419: The paper claims that the previous modeling studies induced seizures by either stimulating pyramidal cells or increasing [K]o, whereas in this study, seizure is induced in the model by stimulating the interneuron. On the surface, this is true. But in reality, the approach adopted by this study is an indirect method of raising extracellular potassium of pyramidal cells. Stimulating interneurons leads to higher [K]o and hence seizures. This is also acknowledged by the authors. However, what is not clear is that why in the network only interneurons would receive strong depolarizing stimulus but not pyramidal cells. It should also be noted that the interplay between interneurons and pyramidal cells during seizures under normal [K]o in a modeling study was first reported by Wei et al. (2014) (PMID: 24671540).

We agree with the Reviewer that the reason for using direct depolarization of the interneuron to initiate SLE could seem unjustified and needed further explanation. Please, see also our answer to Reviewer #2 point 1.

Increased interneuron discharges and decreased PY activity around the time of seizure onset have been shown in SLE induced in vitro by 4-AP (Ziburkus et al., 2006; Lévesque et al., 2016) and bicuculline (Gnatkovsky et al., 2008), in acute and chronic seizure models in vivo (Grasse et al., 2013; Miri et al., 2018; Toyoda et al., 2015) and temporal lobe of epilepsy patients (Elahian et al., 2018). Motivated by this general observation obtained under different experimental conditions, we aimed to test whether fast intraneuronal discharges can indeed trigger SLE in silico and if so, what would be the underlying mechanism. We didn’t aim to rise extracellular potassium of pyramidal cells or increase their discharges by depolarizing stimulus at SLE onset because, as noted above, the transitions to spontaneous seizures that begin with LFV pattern were not associated with increased excitatory activity, but with increased firing of inhibitory interneurons. In the revised manuscript it is explained in ln. 428-430 and the pioneer modeling work by Wei et al., 2014a, which considered an interplay between interneurons and pyramidal cells during seizures, based on the experimental data of Ziburkus et al. (2006) is now quoted in ln. 431-436.

Line 437-439: "This suggests that in our model, a change in the concentrations of either [K^+^]o or [Cl^-^]i was not sufficient to initiate an SLE and that an increase in both is necessary." This seems to be a model-specific effect because several modeling studies have shown that raising [K]o alone can cause the network to enter seizure-like state.

This is a relevant point, and we recognize that our statement was imprecise and should be clarified. In the revised manuscript we modified the sentence mentioned above (ln. 455-457) and we add that our results don’t contradict in vitro experimental observations that elevated [K^+^]_o_ alone is sufficient to induce epileptiform activity (Jensen and Yaari, 1997; Traynelis and Dingledine, 1988). As shown by a bifurcation diagram (Figure 4A and Appendix I – figure 1) an increase in [K^+^]_o_ alone may lead to a transition from a silent state to tonic and burst firing (ln. 460-464).

Line 250: [Na+]I should be [Na+]i.

Thank you, the typo has been corrected.

[Editors’ note: what follows is the authors’ response to the second round of review.]

Essential revisions:We ask that you focus specifically on addressing the following1. Clarify the specific novel contributions of this work in the abstract. The role of Na+, Na+/K^+^ pump on termination and Cl^-^ on seizure initiation have been previously shown.

We clarified the following novel contributions of our work in the abstract:

– We describe the most accurate biophysical model to date replicating electrographic pattern of a typical human focal seizure characterized by low voltage fast activity onset, tonic phase, clonic phase and postictal suppression.

– We report the first in silico demonstration of seizure initiation by inhibitory cells via the initial build-up of extracellular K^+^ due to intense interneuronal firing.

– We identify the ionic mechanisms that may underlie a key feature in seizure dynamics, i.e., progressive slowing down of ictal discharges.

– We predict the exponential scaling of inter-burst intervals, confirmed in the whole guinea pig brain in vitro and human seizure data.

2. The bifurcation analysis that suggests soma and dendrite differences for K^+^ concentration resulting in different regimes is interesting. Could you add to the discussion whether there is any experimental evidence for such significant differences in the concentrations for soma and dendrite?

We appreciate the suggestion to check the existing literature on this subject. Our modeling predictions of different K^+^ concentrations around soma and dendrite during seizures are confirmed by experimental data of simultaneous [K^+^]o recordings in somatic and dendritic layers during hippocampal seizures induced in vivo in anesthetized rats (Somjen and Giacchino, (1985); presented also in Somjen (2004), Figure 8-2). We added this information to the manuscript (ln. 568-572).

3. With reference to the bicuculline model, could you add to the discussion in order to explain your mechanistic reasoning for how this "recruits the interneuronal network"

In the revised manuscript we describe hypotheses regarding the mechanism leading to an increase in interneuronal network activity following a decrease in GABA-A conductance by brief bicuculline application (ln 460-475)

4. As the slowing of the interburst interval is one of the novel aspects of this work, it would be helpful to show that ion changes influence the inter-burst interval variations observed during the course of the seizure. Specifically, could you identify the bursting frequencies from the bifurcation analysis and confirm the role of Na^+^ accumulation in the slowing of burst interval.

We performed further simulations and analyses to demonstrate how a change of different ion types affects seizure termination and inter-burst interval variations (Figure 6 – supplementary figure 1). We conclude that in the model simultaneous changes in [K^+^]_o_ and [Na^+^]_i_ contribute to the slowing of inter-burst interval towards the end of an SLE. Additionally, we consider slowing down scenarios predicted by the bifurcation theory of Izhikevich (2000) and observed in the phenomenological epileptor model of Jirsa et al. (2014). We observe that inter-burst intervals in our biophysical model and in the experimental data don’t follow these predictions and we suggested the reason for this discrepancy (ln. 345-357 and 648-654).

Reviewer #1 (Recommendations for the authors):In this much revised manuscript the authors have improved several features of how ion dynamics were modelled enhancing the validity of their findings. The authors have gone to great lengths to address my concerns. In my opinion the manuscript enhances our understanding of the mechanisms underlying seizure dynamics.

We truly appreciate your positive statements about improvements of the model and the significance of our manuscript.

Novelty[Interneurons driving seizure onset] The authors have generated a thorough model of how intense activity of interneurons can drive K^+^ build-up and the initiation of seizures. This is certainly the most thorough and biophysically realistic model which recapitulates the electrographic feature of human seizures in a satisfying manner.

Thank you for the overall positive evaluation of our manuscript. In the Abstract we emphasized novelty of our study, which shows for the first time in silico that interneuronal firing may initiate seizures via build-up of extracellular potassium.

[Increased Na+/K^+^-pump driving the SLE termination and the post-ictal state]. The authors state that "We show for the first time that seizure termination and postictal state may be generated by the same mechanism mediated by increased activity of the Na+/K^+^-pump. It is an alternative mechanism of the postictal state, which previously has been suggested to depend on potassium undershoot (Krishnan and Bazhenov, 2011)." I disagree with the authors' interpretation and insistence that this is novel. In Krishnan and Bazhenov, they explicitly describe essentially the same mechanism. Their potassium undershoot is caused by increased activity of the Na+/K^+^-pump (the show and state this explicitly)."In our model extracellular potassium decays to baseline after an SLE offset, hence it cannot account for the postictal state." This is also what (Krishnan and Bazhenov, 2011) show (see their Figure 8). I can't see any fundamental difference between your interpretation and theirs, which is certainly satisfying, but not new as far as I can tell.

We thank the Reviewer for the thorough discussion and comments on this topic. We agree that in Krishnan and Bazhenov (2011) postictal state was due to [Na]_i_ accumulation during seizure leading to increased activity of the Na+/K^+^-pump. However, the specific mechanism of reduced postictal excitability is not clear from the paper. In the Abstract, the authors say, “After seizure termination, the extracellular potassium was reduced below baseline, resulting in postictal depression.”. Also, in the Results, a subsection title is: “Below baseline reduction of the extracellular potassium mediates postictal depression state”. On the other hand, at the end of the Introduction they say: “We found that progressive increase in [Na+]i over the course of seizure leads to the changes in the balance of excitatory and inhibitory currents that mediate spontaneous seizure termination and may explain postictal depression state.”. In the Discussion they also say that the reduced postictal excitability was related to both lower than baseline [K^+^]o and increased over baseline [Na+]i (pg. 8880). This view is also presented in their later paper (Krishnan et al., 2015), where they explicitly state that the main cause for termination of seizure and postictal depression state was outward Na/K-pump current. Accordingly, we rephrased the respective paragraph and did not claim the novelty for our finding. Instead, we emphasized that postictal reduction of excitability might be observed both with and without [K^+^]_o_ undershoot and that in our model, reduction in excitability was established through the hyperpolarizing Na/K-pump current (ln. 679-694).

[Exponential IBI distribution]. I agree that the model satisfyingly shows how accounting for ion concentration changes can generate an exponential IBI distribution toward the end of the seizure.

We appreciate your comment. We further extended this part of analysis to demonstrate how individual ion concentration changes influence IBI distribution toward the end of the seizure (Figure 6 —figure supplement 1).

Other responses:I thank the authors for clarifying that in their model bicuculline does not block all of GABAergic inhibition, but is likely a transient perturbation which somewhat counterintuitively recruits the interneuronal network.The authors have gone to great lengths to improve how Cl^-^ flux through GABAaRs and the Cl^-^ leak conductance was modelled. Figure 5 is a wonderful figure (along with Appendix – Figure 3). I am now satisfied that the ion dynamics were modelled in a suitable way which gives me much increased confidence in the authors' findings.

We sincerely appreciate the above comments on the early version of our study as they significantly contributed to improvement of the manuscript.

Reviewer #4 (Recommendations for the authors):In this work, the authors use a computational model to examine the role of ion dynamics in inhibition-mediated TLE seizures. Of significance, the seizure activity precisely matched the experimental data, specifically the time course of the inter-burst interval. The study also replicates previous experimental and computational observations on the role of K^+^ on seizure initiation, Na+, Na+/K^+^ pump on seizure termination. Further, the findings from this work suggest additional contributions of Na+/K^+^ pump to post-ictal depression and K^+^ influence on KCC2 pump promote seizure initiation.While the paper's findings are a significant contribution that emphasizes the importance of ion dynamics on seizure, the novel contribution highlighted by authors seems only as slight variations of previously proposed mechanisms. Nevertheless, the manuscript is definitely worthy of publication, perhaps in a more specialized journal.Few suggestions that could improve the manuscript:1. Bicuculline, a GABA antagonist induced the seizure in the experimental condition, but in the computational model, seizure was induced by increasing the inhibitory neurons' activity. While the increase in inhibitory neuron's activity is observed following bicuculline in the experiment, there is a missing network mechanism that results in this increase of inhibitory neurons activity following the application of bicuculline. It would be more compelling if the authors could identify this mechanism and demonstrate the onset of a seizure by bicuculline in the computational model.

We thank the Reviewer for this relevant comment. Indeed, brief bicuculline application in the isolated guinea pig brain leads to an increase of inhibitory neurons (IN) activity, but the mechanism of this phenomenon is not fully understood. We might hypothesize that in order to fully investigate this effect using a model one should focus on simulating extended interneuronal network with mutual inhibitory interactions, which is unfeasible in the current model. However, in the revised manuscript we describe hypotheses regarding the mechanism leading to an increase in interneuronal network activity following a decrease in GABA-A conductance. Brief bicuculline application likely affects interneuron-interneuron inhibition more than interneuron-principal cell inhibition (Gnatkovsky et al., 2008). Accordingly, reciprocal release of inhibition between IN cells (i.e., disinhibition) could lead to the increased and synchronous firing of inhibitory interneurons and produce preictal spikes generated by interneurons contributing to an increase in extracellular potassium. It would further depolarize the interneuronal network and initiate SLE (de Curtis and Avoli, 2016; Figure 4). In an alternative scenario increased excitability of interneurons could lead to a transition in a bistable IN network, from asynchronous low firing rate mode to synchronous high firing rate mode due to small perturbation (Rich et al. 2020). These hypotheses were added to the Discussion, ln 460-475.

2. It would be very helpful to clarify the specific novel contributions of this work in the abstract. The role of Na+, Na+/K^+^ pump on termination and Cl^-^ on seizure initiation have been previously shown. Also, it would be helpful for authors to note that K^+^ influenced the KCC2 pump time constant in Gonzalez et al., 2018.

We thank the Reviewer for pointing this out. We clarified the novel contributions of our work in the Abstract. Our model generates for the first time an electrographic pattern of a typical human focal seizures consisting of distinct phases, i.e., low voltage fast activity onset, tonic phase, clonic phase and postictal suppression. Our study suggests that interneurons can initiate seizures via an initial build-up of extracellular K^+^ due to intense interneuronal spiking. The model also explains mechanisms that may underlie a key feature in seizure dynamics, i.e., progressive slowing down of ictal discharges. Our model predictions of exponential scaling of inter-burst intervals are confirmed in the whole guinea pig brain in vitro and human seizure data, suggesting that these mechanisms may be preserved across different models and species. We also clarified that in Gonzalez et al., 2018 the KCC2 pump time constant was potassium-dependent (ln 458-459).

3. The bifurcation analysis that suggests soma and dendrite differences for K^+^ concentration resulting in different regimes is interesting. It would be helpful to expand on this finding, and the report is one of the novel contributions. Is there any experimental evidence for such significant differences in the concentrations for soma and dendrite?

We appreciate the suggestion to check existing literature on this subject as we agree that it was not properly discussed. Indeed, during an SLE in the model the increase of [K^+^]_o_ is faster and more pronounced in the somatic vs. dendritic compartment. It is related to activity of the inhibitory neuron, which is located near the soma and also to small longitudinal diffusion between extracellular compartments. As interneuron activity decreases during the course of SLE, the additional contribution to [K^+^]_o_ comes from pyramidal cell firing, The dendritic conductance of voltage-gated K^+^ currents associated with spiking is about 10% of the somatic conductance (Fransen et al., 2002), hence release of K^+^ ions into the dendritic interstitial space is smaller than into somatic interstitial compartment. On the other hand, radial diffusion and glial buffering processes have the same efficiency in both compartments maintaining higher K^+^ near the soma. Our modeling predictions agree with experimental data of simultaneous recordings of [K^+^]o in somatic and dendritic layers during hippocampal seizures induced in vivo in anesthetized rats. During paroxysmal firing induced by electrical stimulation, [K^+^]o in dentate gyrus reached significantly higher levels in cell body layers than in the layers containing dendrites (Somjen and Giacchino, (1985); results reproduced in Somjen (2004), Figure 8-2) We addressed this issue in ln 568-572 of the revised manuscript.

4. It would be helpful to show that ion changes influence the inter-burst interval variations observed during the course of the seizure. Specifically, the authors could identify the bursting frequencies from the bifurcation analysis and confirm the role of Na^+^ accumulation in the slowing of burst interval.

We thank the Reviewer for this comment as it motivated us to perform additional analyses and investigate how a concentration change of K^+^, Na^+^ and Cl^-^ affects inter-burst intervals (IBI). Using slow fast analysis approach as in Figure 4 we varied the concentration of one ion type at a time while holding constant concentrations of the other ions. From the obtained membrane potential traces, we analyzed IBI slowing pattern mediated by changes of each ion type separately. In the analyses, in addition to linear and exponential slowing scenarios, we considered IBI slowing predicted by the bifurcation theory of Izhikevich (2000), who showed that termination of firing may be described by square root or logarithmic relationship of inter-spike intervals. In our analysis, a linear decrease in [K^+^]_o_ led to IBI evolution with exponential characteristics, while a linear increase in [Na^+^]_i_ or decrease in [Cl^-^]_i_ led to SLE termination with logarithmic scaling of IBI as determined by Root Mean Square Error calculated for all four fitted functions (Figure 6 – supplementary figure 1). Yet, the full model, with simultaneously changing K^+^, Na^+^ and Cl^-^ led to exponential slowing. The discrepancy between the results obtained for varying concentration of one ion type at a time and for full SLE in our model and real seizure data suggests that when different factors simultaneously influence seizure termination and the overall slow current change is not necessarily linear, the IBI may not follow slowing predicted by the bifurcation mechanism. We expanded on this finding in the revised manuscript (ln. 345-357 and 648-654).

References:

de Curtis, M., and Avoli, M. (2016). GABAergic networks jump-start focal seizures. Epilepsia, 57(5), 679–687. https://doi.org/10.1111/epi.13370

Gnatkovsky, V., Librizzi, L., Trombin, F., and De Curtis, M. (2008). Fast activity at seizure onset is mediated by inhibitory circuits in the entorhinal cortex in vitro. Annals of Neurology, 64(6), 674–686. https://doi.org/10.1002/ana.21519

Krishnan, G. P., and Bazhenov, M. (2011). Ionic dynamics mediate spontaneous termination of seizures and postictal depression state. Journal of Neuroscience, 31(24), 8870–8882. https://doi.org/10.1523/JNEUROSCI.6200-10.2011

Krishnan, G. P., Filatov, G., Shilnikov, A., and Bazhenov, M. (2015). Electrogenic properties of the Na+/K^+^ ATPase control transitions between normal and pathological brain states. Journal of Neurophysiology, 113(9), 3356–3374. https://doi.org/10.1152/jn.00460.2014

E.M. Izhikevich, Neural excitability, spiking and bursting. International Journal of Bifurcation and Chaos 10(6):1171-1266 (2000)

Jirsa, V. K., Stacey, W. C., Quilichini, P. P., Ivanov, A. I., and Bernard, C. (2014). On the nature of seizure dynamics. Brain, 137(8), 2210–2230. https://doi.org/10.1093/brain/awu133

Rich S, Chameh HM, Rafiee M, Ferguson K, Skinner FK, Valiante TA. Inhibitory Network Bistability Explains Increased Interneuronal Activity Prior to Seizure Onset. Front Neural Circuits. 2020;13:81

Somjen GG, Giacchino JL. Potassium and calcium concentrations in interstitial fluid of hippocampal formation during paroxysmal responses. J Neurophysiol. 1985 Apr;53(4):1098-108. doi: 10.1152/jn.1985.53.4.1098. PMID: 3998794.

G.G Somjen, Ions in the Brain: Normal Function, Seizures, and Stroke (Oxford University Press, USA, 2004).